# Efficient perpendicular magnetization switching by a magnetic spin Hall effect in a noncollinear antiferromagnet

Shuai Hu[1,6], Ding-Fu Shao [2,3,6], Huanglin Yang[1,6], Chang Pan[1,6], Zhenxiao Fu[4,5], Meng Tang[1], Yumeng Yang[4,5] ✉, Weijia Fan[1], Shiming Zhou[1], Evgeny Y. Tsymbal [2] ✉ & Xuepeng Qiu [1] ✉

Current induced spin-orbit torques driven by the conventional spin Hall effect are widely used to manipulate the magnetization. This approach, however, is nondeterministic and inefficient for the switching of magnets with perpendicular magnetic anisotropy that are demanded by the high-density magnetic storage and memory devices. Here, we demonstrate that this limitation can be overcome by exploiting a magnetic spin Hall effect in noncollinear antiferromagnets, such as $Mn_3Sn$. The magnetic group symmetry of $Mn_3Sn$ allows generation of the out-of-plane spin current carrying spin polarization collinear to its direction induced by an in-plane charge current. This spin current drives an out-of-plane anti-damping torque providing the deterministic switching of the perpendicular magnetization of an adjacent Ni/Co multilayer. Due to being odd with respect to time reversal symmetry, the observed magnetic spin Hall effect and the resulting spin-orbit torque can be reversed with reversal of the antiferromagnetic order. Contrary to the conventional spin-orbit torque devices, the demonstrated magnetization switching does not need an external magnetic field and requires much lower current density which is useful for low-power spintronics.

A spin-orbit torque (SOT) provides an effective approach to manipulate the magnetization in spintronic devices[1–5]. In a typical SOT device, an in-plane charge current $J$ generates a spin polarization $p$ via spin-orbit coupling and exerts a torque on the magnetization $m$ of a neighboring ferromagnetic layer. The most efficient SOT driven magnetization switching would require $p$ to be parallel to the easy axis of $m$, so that its associated anti-damping torque $-m \times (m \times p)$ can directly change the effective damping and switch the magnetization of a ferromagnet (FM)[6]. In the conventional spin Hall effect (SHE)[3,4] or the Rashba-Edelstein effect (REE)[1,2], the induced spin polarization $p$, either

in a heavy metal (HM) or at a nonmagnetic metal (NM)/FM interface, is always aligned with the in-plane direction determined by $J \times z$ (where $z$ is normal to the film plane)[7,8]. The associated in-plane anti-damping torque is thus favorable to switch a magnet with in-plane magnetization, while its use for the switching of a magnet with perpendicular magnetization is inefficient and nondeterministic[6]. Although this deficiency could be alleviated by applying an external assisting magnetic field and a high current density (Fig. 1a), these additional requirements would eventually hinder the application of the SOT technique in high-density low-power spintronic devices. To eliminate

[1]Shanghai Key Laboratory of Special Artificial Microstructure Materials and Technology and School of Physics Science and Engineering, Tongji University, Shanghai 200092, China. [2]Department of Physics and Astronomy and Nebraska Center for Materials and Nanoscience, University of Nebraska, Lincoln, NE 68588-0299, USA. [3]Key Laboratory of Materials Physics, Institute of Solid State Physics, HFIPS, Chinese Academy of Sciences, Hefei 230031, China. [4]School of Information Science and Technology, ShanghaiTech University, Shanghai 201210, China. [5]Shanghai Engineering Research Center of Energy Efficient and Custom AI IC, ShanghaiTech University, Shanghai 201210, China. [6]These authors contributed equally: Shuai Hu, Ding-Fu Shao, Huanglin Yang, Chang Pan. ✉e-mail: yangym1@shanghaitech.edu.cn; tsymbal@unl.edu; xpqiu@tongji.edu.cn

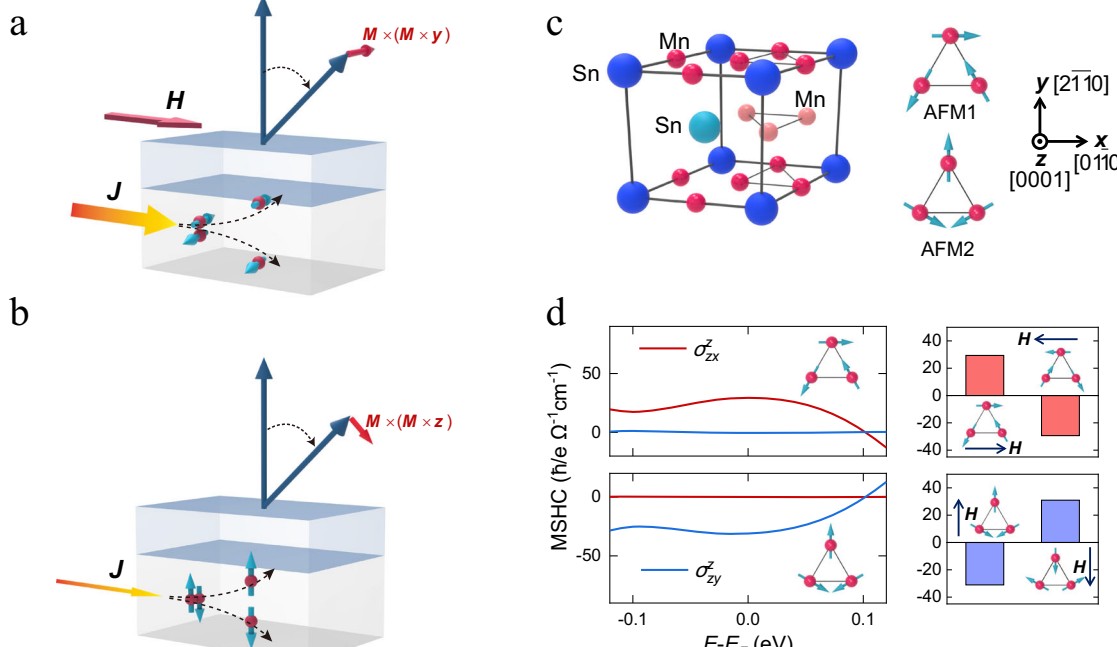

**Fig. 1 | Switching of perpendicular magnetization by damping-like SOTs. a** A schematic of a conventional bilayer SOT device. An in-plane charge current passes along $x$ direction in the bottom spin source layer, generates an out-of-plane spin current with **y**-polarized spin through SHE. This spin current exerts an in-plane damping-like torque $-\boldsymbol{m} \times \boldsymbol{m} \times \boldsymbol{y}$ on the perpendicular magnetization in the top ferromagnetic layer. In this case, a sizable external magnetic field is required for a deterministic switching, and the charge current required is large. **b** A schematic of a bilayer SOT device supporting the out-of-plane anti-damping torque, where the in-plane charge current generates an out-of-plane spin current with **z**-polarized spin. This spin current exerts an out-of-plane anti-damping torque $-\boldsymbol{m} \times \boldsymbol{m} \times \boldsymbol{z}$ on the perpendicular magnetization in the top layer to realize a field-free switching, which

does not require a large charge current. **c** The structure of the noncollinear antiferromagnetic $Mn_3Sn$. The left panel is the side view of the unit cell. The right panel shows the top view of the triangular magnetic alignments of Mn moments within each Mn-Sn Kagome plane. There are two types of the magnetic alignments observed in $Mn_3Sn$, denoted as AFM1 and AFM2. **d** The calculated magnetic spin Hall conductivity $\sigma_{zx}^z$ and $\sigma_{zy}^z$ in $Mn_3Sn$. Left panel shows the $\sigma_{zx}^z$ and $\sigma_{zy}^z$ as a function of energy for AFM1 and AFM2. Right panel shows the sign change of $\sigma_{zx}^z$ and $\sigma_{zy}^z$ at $E_F$ when the magnetic moments in AFM1 and AFM2 are reversed by in-plane magnetic fields. The finite $\sigma_{zx}^z$ and $\sigma_{zy}^z$ indicate $Mn_3Sn$ can be a spin source for the device shown in (**b**) to support an out-of-plane anti-damping torque.

the need of an applied magnetic field for perpendicular SOT switching, an exchange bias induced by an adjacent antiferromagnet (such as IrMn or PtMn)[9,10], a stray field from a neighboring ferromagnetic layer[11], or structure engineering (such as gradient or crystal-growth direction) have been employed[12–15]. The interplay between spin-orbit and spin-transfer torques has been also shown to produce field-free magnetization switching in a magnetic tunnel junction[16].

The symmetry and efficiency restrictions of the SOT devices with conventional in-plane anti-damping torques can be overcome using a more efficient approach exploiting an out-of-plane anti-damping torque $\sim \boldsymbol{m} \times (\boldsymbol{m} \times \boldsymbol{z})$ that directly counteracts the magnetization damping in a perpendicular magnet (Fig. 1b)[17–20]. Therefore, extensive efforts have been devoted to create a $z$-polarized spin current that is capable to produce this kind of SOTs[21]. It has been shown that using a spin source with low crystal symmetry[22,23] and/or an additional magnetic order[23–26] can give rise to the $z$-polarized spin current that is not allowed to appear in the conventional HM spin sources. Alternatively, a proper interface engineering has been reported as a possible mechanism to polarize spins along the $z$-direction through combined actions of spin-orbit filtering, spin precession, and scattering[27–32]. Recently, an out-of-plane anti-damping torque switching of a perpendicular magnet has been experimentally realized in FM/NM/FM trilayers where a $z$-polarized spin current was generated by the FM/NM interface[32], in a heterostructure where the symmetry was lowered by the interface[33], or using a collinear antiferromagnetic (AFM) spin-source layer with a low-symmetry growth direction[34]. While these advances are significant, from the practical perspective, it would be desirable to realize an out-of-plane anti-damping torque switching of a

perpendicular magnet relying entirely on the bulk property of the spin source with a conventional epitaxial growth direction.

Here, we demonstrate that such an out-of-plane anti-damping torque switching can be realized using a noncollinear antiferromagnet $Mn_3Sn$ via the magnetic spin Hall effect[35,36]. We show that this switching is deterministic even in the absence of an applied magnetic field and requires a much lower critical current density than that based on the conventional SHE. Due to being odd with respect to time reversal symmetry, the observed magnetic spin Hall effect and thus the resulting spin-orbit torque can be reversed by an external magnetic field with reversal of the chiral AFM order of $Mn_3Sn$.

## Results and discussion

Bulk $Mn_3Sn$ is a hexagonal compound with the $Ni_3Sn$-type crystal structure[37]. As depicted in Fig. 1c, Mn atoms form Kagome-type lattice planes stacked along the $c$ axis, whereas Sn atoms are located at the center of the Mn-hexagons. The frustration from the triangular geometry of the Kagome lattice results in the chiral alignment of the Mn moments within each plane, leading to a noncollinear AFM order of $Mn_3Sn$ with the Néel temperature ($T_N$) of ~420 K[38–40]. The noncollinear antiferromagnetism in $Mn_3Sn$ results in many interesting properties, such as the large room-temperature anomalous Hall effect[37], the Weyl semimetal phase[41], and the magnetic spin Hall effect (MSHE)[36] relevant to our studies. Different from the conventional SHE being even under time reversal symmetry, the MSHE is odd with respect to this symmetry and hence reversable by flipping the magnetic orders. Resulting from symmetry breaking caused by the noncollinear magnetic structure, the MSHE can be intuitively regarded as a "magnetic" version

of the SHE[35,42,43]. The MSHE gives rise to the unconventional spin currents with distinct symmetries providing useful implications for SOT devices. For example, an out-of-plane spin current with a finite $z$-spin component can be induced by the in-plane charge current (along the $x$ or $y$ direction) in the Mn$_3$Sn (0001) film (Fig. 1c). This spin current is related to the magnetic spin Hall conductivity (MSHC) $\sigma_{zx}^z$ or $\sigma_{zy}^z$ (in the form of $\sigma_{jk}^i$, where $i$, $j$, and $k$ are the spin polarization, spin-current and charge-current directions, respectively). There are two types of the inverse triangular alignment of the magnetic moments in Mn$_3$Sn, denoted as AFM1 and AFM2 in Fig. 1c. The magnetic group symmetry of these alignments supports a small but nonvanishing in-plane net magnetizations along the $x$ ([01$\bar{1}$0]) direction for AFM1 and along the $y$ ([2$\bar{1}\bar{1}$0]) direction for AFM2[38,39]. The presence of this magnetization allows, in particular, the control of AFM domains by a small in-plane magnetic field[38]. This is reflected by our first-principles calculation (Fig. 1d), showing that the $\sigma_{zx}^z$ and $\sigma_{zy}^z$ are finite and reversable for AFM1 and AFM2 of Mn$_3$Sn, respectively (Supplementary Note 1). These sizable MSHCs allow using Mn$_3$Sn as a spin source material to generate the out-of-plane anti-damping SOT and switch the magnetization of an adjacent ferromagnet.

On this basis, we design a MSHE SOT device consisting of a crystalline Mn$_3$Sn (0001) film to generate the MSHE, a [Ni/Co]$_3$ multilayer with perpendicular magnetic anisotropy to control its magnetization, and a thin Cu spacer layer to magnetically decouple Mn$_3$Sn and [Ni/Co]$_3$. Figure 2a schematically shows the designed stack of Mn$_3$Sn(7)/Cu(1)/[Ni(0.4)/Co(0.2)]$_3$/Cu(1)/SiO$_2$(2) (number in the parentheses indicate thickness in nanometers) which is epitaxially prepared by dc magnetron sputtering on cubic MgO (111) substrate. Hall bar devices are subsequently fabricated by standard lithography and etching process for electrical measurements. Resistivities of Mn$_3$Sn and Cu/[Ni/Co]$_3$/Cu are determined to be 367.5 $\mu\Omega$•cm and 45.0 $\mu\Omega$•cm, respectively. The saturation magnetization of the Ni/Co multilayer is measured to be 496 emu/cm$^3$. Through combined characterization techniques of X-ray diffraction, high-resolution transmission electron microscopy, and magneto-transport measurements (Supplementary Note 1, Note 3), the Mn$_3$Sn film is confirmed to have well-defined (0001) crystallographic orientation with a quality comparable to the previously fabricated bulk Mn$_3$Sn crystal[41] and sputtered epitaxial films[44–46], and the whole stack of Mn$_3$Sn(7)/Cu(1)/[Ni(0.4)/Co(0.2)]$_3$ exhibits sharp interfaces and negligible interlayer diffusion. Strong

perpendicular magnetic anisotropy of the [Ni/Co]$_3$ multilayer is confirmed by anomalous Hall effect (AHE) measurements (Supplementary Note 4).

We first verify the existence of the $z$-polarized component in the spin current by measuring the hysteresis loop of the anomalous Hall resistance $R_{AHE}$ versus the out-of-plane magnetic field $H_z$ in the presence of a bias dc current $I$ along the $x$ ([01$\bar{1}$0]) direction. If there was a $z$-polarized spin component, the associated out-of-plane anti-damping torque would cause an abrupt shift of the $R_{AHE}$-$H_z$ hysteresis loop when the torque is sufficiently strong to overcome the intrinsic damping of the Ni/Co multilayer[32]. Indeed, we find no shift of the loop when the amplitude of $I$ is at 4 mA (Fig. 2b), while a sizable positive or negative shift occurs when $I = +16$ mA or $-16$ mA, respectively (Fig. 2c). Here, we define the shift of the $R_{AHE}$-$H_z$ hysteresis loop as $\triangle H_z = H_{center}(I^+) - H_{center}(I^-)$, where $H_{center}(I^\pm) = \frac{[H_r^+(I^\pm) - H_r^-(I^\pm)]}{2}$ is the center of the hysteresis loop determined by the difference of positive and negative magnetization-reversal fields $H_r^\pm(I)$, and $I^\pm$ are positive and negative currents. Figure 2d shows that the threshold current to produce the finite $\Delta H_z$ is around 10 mA, above which the $\Delta H_z$ increases almost linearly with the increase of $I$. This phenomenon cannot emerge in the conventional SOT devices where $\Delta H_z$ can only be generated by a $y$-polarized spin current in the presence of an external magnetic field along the $x$ direction ($H_x$)[32]. Based on the measurements of $\Delta H_z$ with different $H_x$, we estimate the effective SOT fields contributed by the $y$-polarized and $z$-polarized spin currents separately, and derive the sizable spin current conductivities as $|\sigma_{zx}^y| \approx 6.02 \times 10^4$ [($\hbar$/2e) ($\Omega$ m)$^{-1}$] and $|\sigma_{zx}^z| \approx 1.83 \times 10^4$ [($\hbar$/2e) ($\Omega$ m)$^{-1}$] which are comparable to these in prior reports (Supplementary Note 5)[22,23,29,47]. The $|\sigma_{zx}^z/\sigma_{zx}^y|$ ratio of ~30.5% is notable and consistent with the recently reported value for a Mn$_3$Sn single crystal[48], indicating the efficient generation of the $z$-polarized spin current in our Mn$_3$Sn thin films by the MSHE.

The emergence of the $z$-polarized spin current in Mn$_3$Sn allows the realization of the deterministic field-free magnetization switching. Figure 3a shows that the measured AHE resistance as a function of a pulse current $I$ applied along the [01$\bar{1}$0] direction exhibits a hysteretic behavior and a sign change in the absence of applied magnetic field. This behavior reflects the magnetization reversal in the Ni/Co multilayer. We note that in our SOT device, the exchange coupling between the Mn$_3$Sn and Ni/Co multilayer is eliminated by the Cu spacer, and

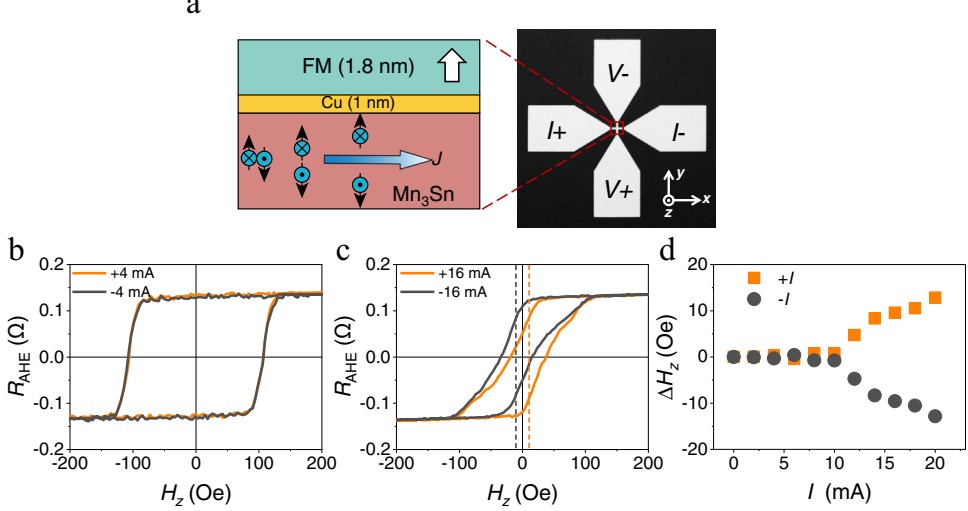

**Fig. 2 | Z-polarized spin current generated by Mn$_3$Sn thin film. a** The schematic of Mn$_3$Sn (7)/Cu (1)/FM (1.8) stack (left) and optical image of the device using for electrical transport measurements (right). The spins with both $\pm y$ and $\pm z$ polarizations generated by bottom Mn$_3$Sn thin film will act on the ferromagnetic layer and induce spin orbit torques simultaneously. **b, c** $R_{AHE}$ vs. $H_z$ curve when the bias currents are $\pm 4$ mA and $\pm 16$ mA. **d** A summary of the shift ($\Delta H_z$) at different bias currents ($I$). The threshold $I$ to cause a shift in AHE curve is about 10 mA. $+I$ will shift the AHE curve to the $+x$ while $-I$ leads to the opposite shift.

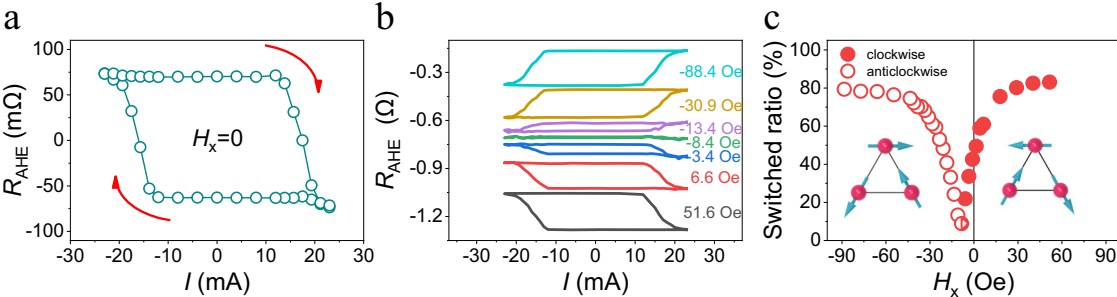

**Fig. 3 | External magnetic field tuning polarity of current induced magnetization switching. a** Current induced magnetization switching with clockwise polarity in the absence of an external magnetic field for the Mn$_3$Sn (7)/Cu (1)/FM (1.8) device. **b** The switching curve under different external magnetic fields from negative to positive. **c** The evolution of the switching polarity and switching ratio under different magnetic fields. Here the direction of magnetic field is in-plane and parallel to the current. The two opposite Mn$_3$Sn domains contribute opposite $z$ spins and thus induce clockwise and anticlockwise switching polarity as indicated as solid and hollow dots.

hence it does not contribute to the magnetization switching (Supplementary Note 6). Instead, the field-free switching is entirely caused by the out-of-plane anti-damping torque. Furthermore, we find that about 60% of the magnetic Ni/Co layer volume switches as estimated from the measured $R_{AHE}$-$I$ loop (Fig. 3a) with respect to the saturation AHE resistance. This partial switching behavior is due to the multi-domain structure of the AFM Mn$_3$Sn film. Due to small magnetic anisotropy and nearly identical energies of AFM1 and AFM2 magnetic orders, the as-grown Mn$_3$Sn films exhibit multiple AFM domains that have a tendency to align randomly in zero field. This gives rise to variation in the magnitude and sign of MSHC $\sigma_{zx}^z$ among different domains, and eventually leads to a reduced effective $\sigma_{zx}^z$. Therefore, the switching ratio is expected to increase if we align the AFM domains. This can be done by applying a finite magnetic field due to a small in-plane magnetization in Mn$_3$Sn[38]. Figure 3b shows the measured $R_{AHE}$-$I$ loops for different magnetic fields $H_x$ along the [01$\bar{1}$0] direction, and Fig. 3c displays the corresponding switching ratio as a function of $H_x$. As seen, by aligning more domains into one configuration, the switching ratio increases as the magnitude of the field increases, with a maximum value of ~80% at a field as small as ~30 Oe. Importantly, the switching polarity is reversed upon reversing the field to opposite direction. All these observations corroborate with the MSHE scenario showing that the MSHC $\sigma_{zx}^z$ is controlled by the reorientation of Mn$_3$Sn domains[44,49]. It is notable that flipping the switching polarity, accompanied by a vanished switching ratio, does not appear at zero field but at a small negative field of about −8 Oe (Fig. 3c). It is likely that some preferred AFM domain orientation is induced by structural reconstructions at interfaces or near defects during the deposition process, which as if there were a small bias field (~ −8 Oe) on Mn$_3$Sn. Similar behavior is found when the current and magnetic field are both along the $y$ ([2$\bar{1}\bar{1}$0]) direction (Supplementary Note 7). We have estimated the Joule heating effect induced by the application of current and found the actual device temperature is ~ 360 K during switching. This temperature is still well below the $T_N$ of Mn$_3$Sn, and therefore the AFM spin texture and the associated spin current largely remain unaffected by the thermal effects (see Supplementary Note 8 for more details). This eliminates Joule heating as the possible origin of the observed effect.

A remaining question is the role of the $z$-polarized spin current in the observed field-free switching: Whether the switching indeed occurs due to an out-of-plane antidamping torque, or the $z$-polarized spin current only generates a symmetry breaking perturbation to assist a conventional SOT switching by the $y$-polarized spin current from the SHE. To answer this question, we have performed macro-spin simulations as described in Supplementary Note 10 and Note 11. We find that for sizable $|\sigma_{zx}^z/\sigma_{zx}^y|$ of 30.5%, as estimated for our Mn$_3$Sn device, the switching dynamics exhibits typical characteristics associated with an out-of-plane antidamping torque. This indicates that

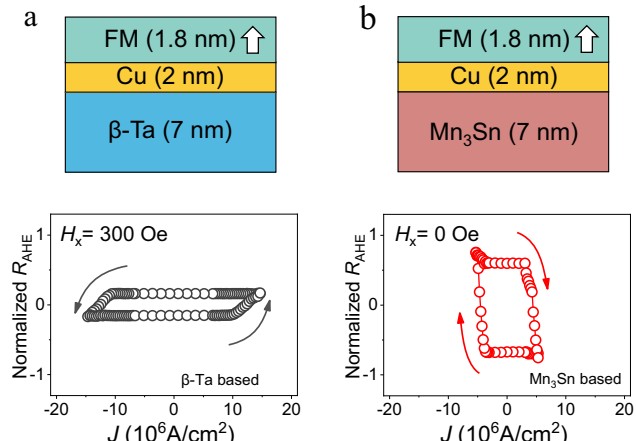

**Fig. 4 | High MSHE based SOT efficiency with the assistance of $z$ spin polarization. a** The conventional SHE based SOT device with a structure of $\beta$-Ta (7)/Cu (2)/FM (1.8) and its maximum current induced magnetization switching curve at the external magnetic field of 300 Oe. **b** The novel MSHE based SOT device with a structure of Mn$_3$Sn (7)/Cu (2)/FM (1.8) and its current induced magnetization switching curve of Mn$_3$Sn based device with the absent of external magnetic field. Note that the current density is calculated by considering the shunting effect of Cu and FM layer in both devices.

the switching of our device occurs through the $z$-polarized spin current driven by the MSHE. On the contrary, for small $|\sigma_{zx}^z/\sigma_{zx}^y|$, we find that the switching is dominated by the $y$-polarized spin current due to the SHE. Importantly, our simulation results indicate that the critical current required for the MSHE-driven switching is much smaller compared to the SHE-driven switching.

Finally, we compare the SOT switching efficiency of the Mn$_3$Sn based MSHE device with a conventional $\beta$-Ta based SHE device. The two devices have the same geometry and thickness of the corresponding layers (Fig. 4a, b). Here, a thicker 2 nm Cu spacer layer is used in both devices as this is the minimum thickness required to achieve good perpendicular anisotropy in a [Ni/Co]$_3$ multilayer for the $\beta$-Ta based device. As a widely explored spin source, $\beta$-Ta has a large charge-spin conversion efficiency to generate the spin current with a conventional in-plane spin polarization[4]. This source, however, requires an assistive magnetic field applied parallel to the charge current for deterministic switching due to the lack of $z$-polarized spin component (Supplementary Note 10 and Note 11). We find that only a small switching ratio of ~17% is realized in this device at the sacrifice of a large current density of $1.3 \times 10^7$ A cm$^{-2}$ and a moderate field of 300 Oe (Fig. 4a). On the contrary, a much larger switching ratio of ~60% is

achieved in the $Mn_3Sn$ based MSHE device even in the absence of external field and under application of a much lower current density of $4.6 \times 10^6$ A $cm^{-2}$ (Fig. 4b). The comparison of SOT switching between the $Mn_3Sn$ and $\beta$-Ta based devices unambiguously proves the superior efficiency of the MSHE induced out-of-plane anti-damping torque for the deterministic switching of perpendicular magnetization.

In conclusion, we have demonstrated the efficient current-induced field-free switching of a ferromagnet with perpendicular magnetization. The magnetization switching is driven by an out-of-plane anti-damping SOT generated by noncollinear antiferromagnet $Mn_3Sn$ resulting from the MSHE. Due to the magnetic spin Hall current being collinear to its spin polarization, this observed mechanism of switching requires much lower critical current density compared to the conventional SHE based SOT devices and allows control through the reorientation of magnetic domains in $Mn_3Sn$. Our findings pinpoint the enormous potential of the MSHE as the spin-torque source to engineer novel energy-efficient spintronic devices.

## Methods

First-principles density functional theory (DFT) calculations were performed using a plane-wave pseudopotential method with the fully-relativistic ultrasoft pseudopotentials[50] implemented in Quantum-ESPRESSO[51]. The exchange and correlation effects were treated within the generalized gradient approximation (GGA)[52]. In the calculations, we used the plane-wave cut-off energy of 35 Ry and a $16 \times 16 \times 16$ k-point mesh in the irreducible Brillouin zone. The experimental lattice parameters $a = 5.689$ Å and $c = 4.522$ Å measured in this work were used in the calculation. All the atomic co-ordinates were relaxed until the force on each atom was less than 0.001 eV/Å. In the calculations, we firstly set the initial magnetic configurations according to the experimentally observed noncollinear AFM states, and then performed full relaxations of the magnetic structure, and the electronic structure without any constraint.

The tight-binding Hamiltonians were obtained from the maximally localized Wannier functions[53] within the Wannier90 code[54] to calculate the MSHC[35]

$$\sigma_{ij}^k = -\frac{e\hbar}{\pi} \int \frac{d^3\vec{k}}{(2\pi)^3} \sum_{n,m} \frac{\Gamma^2 \text{Re}\left(\left\langle n\vec{k}|J_i^k|m\vec{k}\right\rangle \left\langle m\vec{k}|v_j|n\vec{k}\right\rangle\right)}{\left[\left(E_F - E_{n\vec{k}}\right)^2 + \Gamma^2\right]\left[\left(E_F - E_{m\vec{k}}\right)^2 + \Gamma^2\right]} \quad (1)$$

where $J_i^k = \frac{1}{2}\{v_i, s_k\}$ is the spin-current operator, $v_i$ and $s_k$ are velocity and spin operators, respectively, and $i,j,k = x,y,z$. $\Gamma = 50$ meV[35] and a $200 \times 200 \times 200$ k-point mesh were used to evaluate the integral of Eq. (1).

The symmetry determined geometries of conductivity tensors were obtained using the linear response symmetry code[55].

### Sample preparation

All samples were deposited on MgO (111) substrates using DC magnetron sputtering with a base pressure of $5 \times 10^{-8}$ Torr. $Mn_3Sn$ films were deposited with a stoichiometric target at the Ar pressure of $2 \times 10^{-3}$ Torr, the deposition was performed at 100 °C, followed by annealing at 400 °C for 1 h. After the $Mn_3Sn$ film cooled down to the room temperature, Cu and Ni/Co multilayers were then deposited on the $Mn_3Sn$ film with a Cu, Ni, and Co target, respectively. Standard photolithography and Ar ion etching were used to fabricate the 8 μm wide and 35μm long Hall bar.

### Sample characterization

Structural properties of the samples were characterized using a Bruker D8 Discover X-ray diffraction (XRD) system with Cu Kα radiation. The HR-TEM were performed with an electron microscope operated at 200 kV (FEI Titan Themis 200). For the current induced magnetization switching and anomalous Hall loop shift

measurements, current pulses with an 800 μs pulse width were applied. In-between two adjacent writing pulses, the magnetization state of the Ni/Co multilayer was read by current pulses with the same duration as that of the writing pulse but at a much smaller amplitude of 1 mA. The current density is calculated using the standard parallel resistor model which takes into account the shunting effect.

## Data availability

The data that support the findings of this study are available from the corresponding author upon reasonable request.

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

## Acknowledgements

X.Q. and S.Z. acknowledge supports from the National Key R&D Program of China Grant No. 2017YFA0305300, the National Natural Science Foundation of China Grant Nos. 52022069, 11974260, 11674246, 11874283, 51801152, and 11774064, Natural Science Foundation of Shanghai Grant No. 19ZR1478700, and the Fundamental Research Funds for the Central Universities. Y.Y. acknowledges support from National Natural Science Foundation of China Grant No. 62074099. E.Y.T. acknowledges support from the EPSCoR RII Track-1 program (NSF Award OIA-2044049).

## Author contributions

S.H. and X.Q. conceived and designed the experiment. S.H., H.Y. and C.P. fabricated the samples and performed the measurements. S.H., D.F.S., H.Y., C.P., M.T., Y.Y., W.F., S.Z. and X.Q. analyzed and discussed the experiment results. D.F.S., E.Y.T. analyzed the data and performed the numerical calculation. Y.Y., D.F.S. and Z.F. performed the simulation. S.H., D.F.S., Y.Y. and X.Q. wrote the manuscript with contributions from all the authors. All authors discussed the results and commented on the manuscript.

## Competing interests

The authors declare no competing interests.
