## [Peer Review File · Nature Communications]

Efficient perpendicular magnetization switching by a magnetic spin Hall effect in a noncollinear antiferromagnetThis manuscript has been previously reviewed at another journal that is not operating a transparent peer review scheme. This document only contains reviewer comments and rebuttal letters for versions considered at *Nature Communications*.

REVIEWER COMMENTS

Reviewer #1 (Remarks to the Author):

This work reports an interesting study concerning the effects of special Mn₃Sn AFM texture on the field-free SOT switching with remarkable current efficiency. The spintronics effects in Mn₃Sn are recently of wide interests, and this work took the advantage of magnetic spin Hall effects to apply for the SOT field-free switching. The results are interesting but several critical issues listed below need to be clearly addressed.

1. In Fig. 2, to prove the presence of Z-polarized spin current generated by Mn₃Sn thin film, authors apply a large dc current. When the current reaches 16 mA, a shift of AHE curve was observed.

The AHE curve is very different from one with 4 mA, strongly suggest the joule heating effects. Since the Neel temperature of Mn₃Sn is only 420 K (for thin films, it can be even lower), the temperature rise during the dc current applied may significantly disturb the spin texture of Mn₃Sn; therefore, the ideal spin configuration of Mn₃Sn may not be valid to claim the Z-polarized spin current. Although the SOT writing was performed by using pulse, the disturbance of spin texture of Mn₃Sn still existed. In addition to Mn₃Sn spin texture, other origins, for example, interfaces may also give the possible Z-polarized spin current. Authors need to comment on the disturbance of spin texture by current and exclude other possible sources of Z-polarized spin current.

2. In Fig. 3b, the corresponding switching ratio is increased by increasing H_x. Authors explain this results by the increased aligned Mn₃Sn domains. However, when H_x is increased, the symmetry breaking in FM is also increased so FM is easy to be switched. Authors need to give strong evidence to demonstrate that the Mn₃Sn domains are indeed changed.

3. In Fig. 4a and Fig. 4b, is the switching current density gained by the total current injection or actually considering the real shunting in Mn₃Sn and β -Ta. I suppose it should be the latter case for a fair comparison because the resistance of Mn₃Sn and β -Ta should be significantly different.

4. Following the previous question, it is better to provide the H_k of the two investigated samples for the comparison of critical current density. The different H_k in the two systems would also affect the critical current.

5. Furthermore, the spin mixing conductance at β -Ta/Cu and Mn₃Sn/Cu interface may be different, hence leading to a different degree of spin accumulation for SOT. For most SHE-based cases, the insertion of Cu is not necessary, which may be also the reason for the large difference on critical current because the two investigated systems underwent different SOT-switching mechanism, i.e. with and without z-component of spin current. The possibilities mentioned above should be discussed before the authors give the conclusion.

6. From SFig. 3, it seems when the current (I) is transverse to the magnetic field (H), it results in a larger magnetoconductivity although the sign of magnetoconductivity is opposite to the parallel case. Is that possible to perform the SOT switching with varying H_y? In principle, it can still show the SOT switching but the polarity may be opposite to the regular H_x-assisted switching.

7. Following the previous question, the calculation also shows the higher magnetoconductivity when the current (I) is transverse to the magnetic field (H). One can be anticipated that the more spin current can be produced under this geometry. Does it also result in the reduced critical current density?

Reviewer #2 (Remarks to the Author):

This paper investigates the magnetization switching in CoNi perpendicular ferromagnet by the z-directional polarized spin current generated from Mn₃Sn antiferromagnet via magnetic spin Hall effect (MSHE). Magnetization switching by spin-orbit torque without external magnetic field is a challenging topic from the viewpoints of not only fundamental physics but also practical applications. The previous works had mainly focused on technical assistances, such as the exchange bias from antiferromagnets, the tilted magnetic anisotropy, and so on, to achieve the magnetization switching caused by spin Hall effect (SHE). Recently, on the other hand, other proposals focusing on the spin-current generation from ferromagnets have been reported, where breaking time-reversal symmetry due to the presence of the ferromagnet enables us to manipulate the direction of the spin polarization in a different way from SHE. These previous works are correctly referred in the introduction. The other proposals have focused on the spin-current generation from antiferromagnet, as done in this work. Using Mn₃Sn as an antiferromagnetic spin source, the present work reports the magnetization switching in CoNi ferromagnet.

This work is well done, and the paper is well written. I think the paper will be of interest in spintronics community. Simultaneously, however, the work lacks impact to satisfy high publication standard of Nature Communications; see my comments below. My recommendation is to publish this work from Communications Physics after the authors address the following issues.

1. I could not agree with the authors to use "efficient" in their title because its definition is unclear. From the viewpoint of practical applications, which is emphasized in the introduction, the charge and spin conductivities are key quantities to quantify the efficiency because it determines the power consumption of magnetic devices. In this sense, it is highly desirable to evaluate the values of " σ^z_{zx} " and " σ^z_{zy} " experimentally and compare them with the other systems mentioned in the introduction. For example, it was recently reported that the conductivity of spin current CoNiB alloy is on the order of $10^3 \text{ Ohm}^{-1} \text{ cm}^{-1}$; see Phys. Rev. Applied 14, 064056 (2020). Are the values of " σ^z_{zx} " and " σ^z_{zy} " of the present work, for example, larger than this previous work? I do not prefer to use the spin Hall angle as an efficiency because the value can be high by using materials with high longitudinal resistivity. In addition, as I wrote above, the power consumption is determined by the conductivity, and therefore, the spin Hall angle is not a direct measure to discuss the efficiency.

As far as I know, the conductivity of Mn₃Sn is not so high compared with the metallic ferromagnets, and therefore, the bulk and interfacial generations of spin current from ferromagnets, cited as Refs. 23-32, are more "efficient" than MSHE, in my opinion. To guarantee the impact of this work and use the word "efficient" in title, I recommend the authors to specify the value of the spin conductivity and show that the value is higher than the previous proposals.

2. The switching experiment lacks impacts due to the following two reasons. First, the switching probability ("portion" the authors wrote) is much lower than 100%, as can be seen in, for example, Fig. 3c. I think that the switching probability at zero H_x reflects that purely driven by MSHE. Unfortunately, the switching probability remains approximately 50%, which is too low to apply magnetic devices. The low value of the switching probability makes the superiority of MSHE, compared to the other methods mentioned in the introduction, unclear. Second, I think that not only MSHE but also SHE contributes to the switching at finite H_x . Therefore, it is necessary to evaluate the contributions from MSHE and SHE separately for guaranteeing the presence of MSHE and its importance. Again, it is necessary to evaluate the conductivity of SHE.

In summary, the work is well done and reliable. Therefore, the paper will attract readers in Nature journals. However, the work lacks enough impact to guarantee the publication from Nature Communications. Accordingly, Communications Physics would be suitable for publication.

Minor comments.

1. I recommend the authors to check their presentations of figures. For example, it is very hard to read the values of the magnetic field written in Fig. 3b because of low resolution and small font size. The unit of the conductivity in the vertical axis of Fig. S3, as well the "degree" in the horizontal axis of Fig. S3c is not shown correctly (I had downloaded the pdf version of Supplementary Information). Compared with Fig. 1d, I assumed that the unit is " $\text{Ohm}^{-1} \text{cm}^{-1}$ " in the horizontal axis of Fig. S3; is it correct?

2. It would be preferable to write the pulse width of the current for switching explicitly because fast magnetization switching is required in practical applications.

Reviewer #3 (Remarks to the Author):

This paper reports:

- (a) The detection of an out-of-plane antidamping torque generated by the non-collinear antiferromagnet Mn₃Sn based on current-dependent changes in the out-of-plane coercive magnetic field (Fig. 2)**
- (b) Partial switching of a perpendicularly-magnetized Ni/Co multilayer with microns-scale lateral dimensions by spin-orbit torque in zero applied magnetic field (Fig. 3a)**
- (c) A change in the symmetry and degree of completeness in this current-induced switching as a function of the magnitude and direction of the external magnetic field, that can be associated with the rearrangement of antiferromagnetic domains (Fig. 3b,c)**
- (d) A comparison between spin-orbit switching of a Ni/Co multilayer by Ta and Mn₃Sn in samples with microns-scale lateral dimensions, showing that the Mn₃Sn allows switching at a lower current density.**

The paper is well-written, and I find all of the results to be credible and well supported by the data shown. However, in my opinion, the results shown so far do not move the field forward sufficiently to merit publication in Nature Communications. I suggest that paper be reconsidered after the authors have performed additional measurements.

Major comments:

1. Measurement of an out-of-plane antidamping torque from a non-collinear magnet is not new. In addition to the papers cited in the manuscript under review, there are also results by Nan et al. (already in Nature Commun. 11:4671 (2020)) that the manuscript under review does not cite. That previous paper provided a quantitative measurement of the spin torque conductivity for the antidamping spin-orbit torque, which the paper under review does not provide. I do find it interesting that Mn₃Sn also provides an out-of-plane antidamping torque, but to move the field forward it is important to know how the strength of this torque compares quantitatively to previous measurements. The quantitative strength of the ordinary in-plane anti-damping spin-orbit torque should also be reported.

2. Because all of the measurements are performed on samples with microns-scale lateral dimensions, the magnetic switching occurs by a process of domain wall motion. Under the influence of an applied current, this process is thermally-activated so that the primary effect of the current is heating rather than spin-orbit torque (note that the coercive magnetic field is reduced for both directions of current in Fig. 2(c) compared to

Fig. 2(b), with just a small asymmetry due to spin torque as a function of the sign of the current. I realize that this is a common regime for study within the literature, but despite this fact it is not really a useful regime for obtaining any truly quantitative information about spin-orbit torques, because the thermally-activated domain-wall depinning process is poorly-controlled and difficult to model quantitatively. It is also not a useful regime for drawing any conclusions about potential applications, because practical devices will require devices much smaller in scale than 100 nm where switching is not dominated by domain-wall motion. The current densities needed to drive switching on sub-100-nm scale devices can be a factor of 50 or more greater than for devices on the many-microns scale because the switching occurs by completely different mechanisms in these two regimes (see C. Zhang et al., Appl. Phys. Lett. 107, 012401 (2015)). To show that the out-of-plane antidamping torque is strong enough for practical applications, I suggest that the authors should show that it can assist switching for devices close to the 100 nm scale or below, and not just for microns-scale devices. If the data in Fig. 2d really do indicate a full anti-damping transition driven by the current, it appears that the current densities needed to switch 100 nm scale devices might be achievable.

3. Another important question that would move the field forward, but the manuscript under review does not address, is whether in the switching they observe the out-of-plane antidamping torque merely provides some symmetry breaking so that the main driver of the switching is still the ordinary (inefficient) in-plane anti-damping torque with a positive overall effective damping, or whether the out-of-plane antidamping torque is strong enough to drive the more-efficient anti-damping switching mechanism.

More minor comments:

4. The paper does not appear to provide all of the materials parameters that readers will want to know for a full analysis of the results, for example the resistivities of all of the various layers and the average magnetic moment per unit volume of the Ni/Co multilayer.

5. I would have found it useful to understand what is the arrangement of the spins that allow a small nonzero total magnetic moment in the Mn₃Sn. If the lower two spins shown in the diagram for AFM1 in Fig. 1(c) point exactly along the sides of the equilateral triangle as shown, the net magnetization will be zero. I realize that the canting away from this configuration will be small, but I suggest that the authors indicate this canting somehow.

6. I suggest that the authors state explicitly how they define the quantity ΔH_z graphed in Fig. 2(d). For a given value of applied current, is it the difference of the coercive fields for the magnet originally in the up configuration and the down configuration? I do not believe this is ever stated explicitly.

On a positive note, it is interesting and exciting that the authors observe signs of an out-of-plane antidamping torque from a new material (Mn₃Sn), and I find it really neat that they are able to control the overall effectiveness of this torque by applying a magnetic field to change the arrangement of antiferromagnetic domains. However, to merit publication in Nature Communications it is my opinion that the work should move the field forward more significantly than this. I suggest the authors perform additional measurements to address the questions noted in my comments 1-3 above. If they can address a significant portion of these comments I would likely recommend the paper for publication in Nature Communications.

Response to Reviewers

Reviewer #1:

General Comment: This work reports an interesting study concerning the effects of special Mn₃Sn AFM texture on the field-free SOT switching with remarkable current efficiency. The spintronics effects in Mn₃Sn are recently of wide interests, and this work took the advantage of magnetic spin Hall effects to apply for the SOT field-free switching. The results are interesting but several critical issues listed below need to be clearly addressed.

Response: We are pleased that the reviewer finds our work “interesting” and appreciate his/her insightful and helpful suggestions. Below, we respond point-to-point to the reviewer’s comments.

Comment #1: In Fig. 2, to prove the presence of Z-polarized spin current generated by Mn₃Sn thin film, authors apply a large dc current. When the current reaches 16 mA, a shift of AHE curve was observed. The AHE curve is very different from one with 4 mA, strongly suggest the joule heating effects. Since the Neel temperature of Mn₃Sn is only 420 K (for thin films, it can be even lower), the temperature rise during the dc current applied may significantly disturb the spin texture of Mn₃Sn; therefore, the ideal spin configuration of Mn₃Sn may not be valid to claim the Z-polarized spin current. Although the SOT writing was performed by using pulse, the disturbance of spin texture of Mn₃Sn still existed. In addition to Mn₃Sn spin texture, other origins, for example, interfaces may also give the possible Z-polarized spin current. Authors need to comment on the disturbance of spin texture by current and exclude other possible sources of Z-polarized spin current.

Response: The reviewer is correct that the SOT measurement requires a large current injection and involves a non-negligible Joule heating effect. Following the reviewer’s suggestion, we have determined the device actual temperature by calibrating the device resistance under pulse current injection (Fig. R1a) against the temperature dependence of device resistance (Fig. R1b). The actual device temperatures were estimated to be ~340 K for the maximum current used in Fig. 2c and ~360 K for the critical switching current used in Fig. 3a, which are well below the $T_N = 420$ K of Mn₃Sn. This indicates that the phenomena we observed in this work are indeed due to the noncollinear antiferromagnetism of Mn₃Sn.

Figure R1: **a** Resistance of Mn_3Sn (7)/Cu (1)/FM (1.8) device vs. pulse current amplitude (pulse width = 200 μs). **b** Linear fitting curve for the temperature dependence of device resistance. **c** Determination of device temperature with different pulse amplitudes.

The reviewer is correct that other origins, such as the interface scattering, may support the z -polarized spin current. However, we believe that the interfaces do not play a decisive role in our study. There are two main mechanisms reported for a z -polarized spin current to be generated by interfaces. The first mechanism is due to spin-orbit coupling at a low symmetry NM/FM interface, supporting a localized nonequilibrium z -spin polarization as reported in Py/WTe₂ system (MacNeill *et al.*, Nat. Phys. 13, 300 (2016)) and CuPt/CoPt system (Liu *et al.*, Nat. Nanotechnol. 16, 277, (2021)). This effect is not expected to be sizable in our devices, due to spin-orbit coupling, in both Cu and FM layers, being very weak. In addition, the spin polarization generated by this effect cannot be reversed by a magnetic field, thus cannot explain the evolution of switching polarity as we observed in Fig. 3.

The second mechanism is the spin-orbit precession mechanism in a FM/NM/FM trilayer (Amin *et al.*, Phys. Rev. Lett. 121, 136805 (2018); Baek *et al.*, Nat. Mater. 17, 509 (2018)), where the z -polarized spin current is generated by the spin precession of an in-plane x -polarized longitudinal current due to the interfacial spin-orbit field. This mechanism, in principle, can appear in our system, since Mn_3Sn possesses a nonvanishing in-plane net magnetic moment and hence can generate an in-plane x -polarized longitudinal current. However, since the generated out-of-plane spin current in this mechanism is due to the interfacial transmission and reflection (Amin *et al.*, Phys. Rev. Lett. 121, 136805 (2018)), this current is not expected to be as strong as that generated by a bulk effect such as the MSHE.

In addition, we have performed additional measurements to quantitatively estimate the spin conductivities, as shown in Supplementary Information S5. We found that $|\sigma_{zx}^z/\sigma_{zx}^y|$ is as large as ~30.5%, which is consistent with the recent reports for a Mn_3Sn single crystal (Kondou *et al.*, Nat. Commun. 12, 6491 (2021)). This indicates that the efficient generation of the z -polarized spin current by Mn_3Sn has the bulk origin.

We have clarified the Joule heating effect and estimated the actual device temperature in the revised manuscript (first paragraph on page 6). The details of calibrating Joule heating have been provided in Supplementary Information S8:

“We have estimated the Joule heating effect induced by the application of current and found the actual device temperature is ~ 360 K during switching (Supplementary Information S8). This temperature is still well below the T_N of Mn_3Sn , indicating that the observed phenomena are originated from the noncollinear antiferromagnetism of Mn_3Sn .”

The discussion regarding the interface origin of the z -polarized spin current has been added in Supplementary Information S5:

“We note that in addition to the MSHE, there might be other contributions to the z -polarized spin current in the Mn_3Sn -based SOT, resulting from interfaces^{8,10,13}. However, we expect these contributions play a minor role in the field-free switching we observe. The spin currents driven by interface-related mechanisms are either independent on the magnetic order parameter by symmetry^{10,13}, or not strong enough compared to that generated by the bulk effect such as MSHE. For example, it has been suggested that the nonequilibrium z -spin polarization can be generated by spin-orbit coupling at low symmetry normal metal (NM)/ferromagnet (FM) interfaces, as reported for Py/WTe_2 ¹³ and $CuPt/CoPt$ ¹⁰ systems. This effect is not expected to be sizable in our devices, since spin-orbit coupling in both Cu and FM layers is very weak. In addition, the spin polarization generated by this effect cannot be reversed by a magnetic field, and thus cannot explain the change of the switching polarity as shown in Fig. 3 of the main text. The z -polarized spin current due to spin-orbit precession as suggested for a $FM/NM/FM$ trilayer^{8,14}, in principle, can appear in our system, since Mn_3Sn can host a nonvanishing in-plane net magnetic moment and hence can be considered to be a weak ferromagnet. However, the generated out-of-plane spin current in this mechanism is due to the

interfacial transmission and reflection¹⁴, and hence is not expected to be as strong as that generated by a bulk effect such as the MSHE in present study. A good agreement between the z-polarized spin current value in our measurements and that in the recent report for a Mn₃Sn single crystal¹² indicates that the SOT switching we observed has largely the bulk origin.”

Comment #2: In Fig. 3b, the corresponding switching ratio is increased by increasing H_x . Authors explain this results by the increased aligned Mn₃Sn domains. However, when H_x is increased, the symmetry breaking in FM is also increased so FM is easy to be switched. Authors need to give strong evidence to demonstrate that the Mn₃Sn domains are indeed changed.

Response: Following the reviewer’s suggestion, we have measured the Mn₃Sn magnetization by a SQUID magnetometer. As shown in Fig. R2, Mn₃Sn thin film has a very soft magnetic hysteresis behaviour at room temperature, indicating that Mn₃Sn domains can be manipulated by an external magnetic field. Moreover, with the thermal activation by Joule heating, one can expect that the domains may be reoriented even in a field much smaller than the coercive field at room temperature. As a result, in our Mn₃Sn device, a large AFM domain portion can be switched by a small magnetic field H_x . On the contrary, the H_x field required for symmetry breaking in a conventional SOT device is usually sizable.

Importantly, unlike in the conventional SOT device, the switching polarity in our Mn₃Sn based SOT device is reversed at a finite H_x field. This clearly points to the reversal of AFM domains in our Mn₃Sn device. Therefore, we believe that the observed switching enhancement by the H_x field and field-free SOT switching is understood adequately by the MSHE scenario with H_x induced domain reorientations.

Figure R2: Magnetic hysteresis loop of a 7 nm Mn₃Sn film. The inset shows the amplified loop data at small magnetic fields.

In this revision, we have added discussion of the magnetic properties of Mn₃Sn thin films in Supplementary Information S2 and included Fig. R2 as Fig. S3.

“We also characterize the magnetic properties of as-deposited film. The as-deposited Mn₃Sn film exhibits a very soft magnetic hysteresis behaviour at room temperature (Fig. S3), indicating that its domains can be manipulated by an external magnetic field. With the thermal activation by Joule heating, when the film is patterned into a Hall bar device, the domains can be reoriented even in a field much smaller than the coercive field at room temperature.”

Comment #3: In Fig. 4a and Fig. 4b, is the switching current density gained by the total current injection or actually considering the real shunting in Mn₃Sn and β -Ta. I suppose it should be the latter case for a fair comparison because the resistance of Mn₃Sn and β -Ta should be significantly different.

Response: The reviewer’s understanding is correct. The switching current density in Fig. 4a,b of the original manuscript is calculated by considering the current shunting effect between layers. In our experiment, we grew the Mn₃Sn and Ta single layers and measured their resistivities to be 367.5 and 167.9 $\mu\Omega\cdot\text{cm}$, respectively. Then, we grew a Cu/CoNi/Cu multilayer, which includes the rest layers in the stack, and measured its resistivity to be 45.0 $\mu\Omega\cdot\text{cm}$. From these resistivity values, the current flowing within the Mn₃Sn or Ta layers was calculated using a parallel resistor model.

Following this comment, we have clarified how the current density was calculated in our revised manuscript (Methods section):

“The current density is calculated using the standard parallel resistor model which takes into account the shunting effect.”

The Mn₃Sn resistivity value has been quoted in the second paragraph on page 4.

“Resistivities of Mn₃Sn and Cu/[Ni/Co]₃/Cu are determined to be 367.5 μΩ-cm and 45.0 μΩ-cm, respectively.”

Comment #4: Following the previous question, it is better to provide the H_k of the two investigated samples for the comparison of critical current density. The different H_k in the two systems would also affect the critical current.

Response: We agree with the reviewer that the H_k is an important parameter, which can affect the SOT switching critical current density. The H_k of the Mn₃Sn and Ta samples with 2 nm Cu have been determined to be 2487 Oe and 1973 Oe through transport measurements as in our previous work (Yang *et al.* Phys. Rev. B 102, 024427 (2020)). This clearly excludes the possibility that the reduction of the switching current in Mn₃Sn is due to the decrease of H_k .

Following this comment, we have provided these H_k values in Supplementary Information S4:

“The perpendicular magnetic anisotropy field H_k has been also determined to be 1727 Oe, 2487 Oe and 1973 Oe for the Mn₃Sn (7)/Cu (1)/FM, Mn₃Sn (7)/Cu (2)/FM and Ta (7)/Cu (2)/FM samples, respectively.”

Comment #5: Furthermore, the spin mixing conductance at β-Ta/Cu and Mn₃Sn/Cu interface may be different, hence leading to a different degree of spin accumulation for SOT. For most SHE-based cases, the insertion of Cu is not necessary, which may be also the reason for the large difference on critical current because the two investigated systems underwent different SOT-switching mechanism, i.e. with and without z-component of spin current. The possibilities mentioned above should be discussed before the authors give the conclusion.

Response: We thank the reviewer for this valuable suggestion. There are three main reasons for using the Cu interlayer in this work. First, the Cu layer effectively blocks the exchange bias effect between Mn₃Sn and FM (Supplementary Information S6), which on its own to stimulate field-free SOT

switching, thus complicating the study (Oh *et al.* Nat. Nanotechnol. 11, 878 (2016); Fukami *et al.* Nat. Mat. 15, 535 (2016)). Second, the Cu layer is necessary to induce perpendicular magnetic anisotropy of the Co-Ni layer. Third, due to the small SOC strength, Cu allows an efficient spin transport.

The reviewer is correct that different spin mixing conductances at the β -Ta/Cu and Mn₃Sn/Cu interfaces could influence the SOT switching (Zhang *et al.*, Nature Physics 11, 496 (2015)), and thus could affect our conclusions. To respond to this valid concern of the reviewer, we have prepared a Ta-based SOT device by adding a Ta layer directly on top of the Co-Ni ferromagnet, and found that even in the absence of the Cu spacer between the Ta and FM layers, the switching current of 9.2×10^6 A/cm² in this device is still significantly larger than that of the Mn₃Sn-based device with the Cu spacer (Supplementary S8). We thus conclude that the spin mixing conductance does not affect our conclusion, and our results unambiguously prove the efficiency of the SOT switching induced by the out-of-plane anti-damping torque due to the MSHE in our Mn₃Sn based device compared to that in the conventional SOT devices.

Figure R3: SOT switching curve for the Ta (3 nm)/Cu (2 nm)/Co-Ni (1.8 nm)/Ta (4 nm)/TaO_x (2 nm) device. A 300 Oe assisted field is applied along current direction.

We have included Fig. R3 in Supplementary Information S8 as Fig. S12 and added the following text there:

“To elucidate the influence of the inserted Cu layer on the SOT device performance, we have prepared a Ta-based SOT device by adding a Ta layer directly on top of the ferromagnet. The stack structure is Ta (3 nm)/Cu (2 nm)/Co-Ni (1.8 nm)/Ta (4 nm)/TaO_x (2 nm) and a good perpendicular magnetic anisotropy with $H_K = 1850$ Oe is ensured by the Cu layer below the FM layer. As seen from Figure S12, the switching current of 9.2×10^6 A/cm² at 300 K in this device is still significantly larger than that in the Mn₃Sn-based device with the Cu spacer (Fig. 4b). These results eliminate a possible concern that the different spin mixing conductance at the β -Ta/Cu and Mn₃Sn/Cu interfaces is responsible for a much more efficient performance of our Mn₃Sn based device compared to the conventional SOT devices.”

Comment #6: From SFig. 3, it seems when the current (I) is transverse to the magnetic field (H), it results in a larger magnetoconductivity although the sign of magnetoconductivity is opposite to the parallel case. Is that possible to perform the SOT switching with varying H_y ? In principle, it can still show the SOT switching but the polarity may be opposite to the regular H_x -assisted switching.

Response: We thank the reviewer for this suggestion. As we mentioned in the manuscript, Mn₃Sn has two different magnetic configurations, AFM1 and AFM2. The magnetic group symmetry of these configurations supports a small but nonvanishing in-plane net magnetization along the x ([01 $\bar{1}$ 0]) direction for AFM1 and along the y ([2 $\bar{1}$ $\bar{1}$ 0]) direction for AFM2. Therefore, one can expect that the magnetic domains will be gradually reoriented to AFM2 configuration by a magnetic field along the y direction (H_y). Since the spin Hall conductivity σ_{zx}^z , which is related to the field-free switching, is finite in the AFM1 configuration but absent in the AFM2 configuration by symmetry (see Table S1), the z -polarized spin current will decrease to zero as H_y favors the AFM2 domains but suppresses the AFM1 domains. Therefore, we expect that H_y will suppress the switching but won't influence the polarity, since it only reduces the out-of-plane z -polarized spin current but does not reverse its spin polarization. In order to prove this, we have performed additional SOT measurements with current along the x ([01 $\bar{1}$ 0]) direction and the magnetic field along the y ([2 $\bar{1}$ $\bar{1}$ 0]) direction, as shown in Fig. R4. As expected, the switching was indeed suppressed by the application of H_y , and the polarity was well maintained.

Figure R4: **a** The switching curve under different external magnetic fields along the $[2\bar{1}\bar{1}0]$ direction with I being along the $[01\bar{1}0]$ direction. **b** The switching ratio indicating the portion of the switched AMF domains as a function of magnetic field H_y .

In this revision, we have included these additional measurements and added the above discussion to Supplementary Information S7 with Fig. R4 included as Fig. S9.

“We have also measured the SOT switching when the current is applied along the x ($[01\bar{1}0]$) direction and the external magnetic field along the y ($[2\bar{1}\bar{1}0]$) direction. Due to the magnetic spin Hall conductivity σ_{zx}^z , which is related to the field-free switching of the device, being finite in the AFM1 configuration but absent in the AFM2 configuration by symmetry (Table S1), an out-of-plane spin current is expected to gradually decrease to zero in this case as H_y favors the AFM2 domains and suppresses the AFM1 domains. Therefore, we expect that H_y will suppress the switching but won't influence the polarity, since it only reduces the z -polarized spin current but does not reverse its spin polarization. This is indeed has been confirmed in our measurements, as seen from Fig. S9.”

Comment #7: Following the previous question, the calculation also shows the higher magnetoconductivity when the current (I) is transverse to the magnetic field (H). One can be anticipated that the more spin current can be produced under this geometry. Does it also result in the reduced critical current density?

Response: As mentioned in our response to Comment#6, the application of H_y reorients Mn_3Sn

domains toward the AFM2 configuration which does not support σ_{zx}^z . Therefore, it cannot reduce the critical current density, as has been proved by our additional measurements. The critical switching current at different H_x and H_y values are summarized in Fig. R5.

Figure R5: The critical switching current under different H_x (along the $[01\bar{1}0]$ direction) and H_y (along the $[2\bar{1}\bar{1}0]$ direction). The current is applied along the $[01\bar{1}0]$ direction.

Reviewer #2

General Comment: This paper investigates the magnetization switching in CoNi perpendicular ferromagnet by the z-directional polarized spin current generated from Mn₃Sn antiferromagnet via magnetic spin Hall effect (MSHE). Magnetization switching by spin-orbit torque without external magnetic field is a challenging topic from the viewpoints of not only fundamental physics but also practical applications. The previous works had mainly focused on technical assistances, such as the exchange bias from antiferromagnets, the tilted magnetic anisotropy, and so on, to achieve the magnetization switching caused by spin Hall effect (SHE). Recently, on the other hand, other proposals focusing on the spin-current generation from ferromagnets have been reported, where breaking time-reversal symmetry due to the presence of the ferromagnet enables us to manipulate the direction of the spin polarization in a different way from SHE. These previous works are correctly referred in the introduction. The other proposals have focused on the spin-current generation from antiferromagnet, as done in this work. Using Mn₃Sn as an antiferromagnetic spin source, the present work reports the magnetization switching in CoNi ferromagnet.

This work is well done, and the paper is well written. I think the paper will be of interest in spintronics community. Simultaneously, however, the work lacks impact to satisfy high publication standard of Nature Communications; see my comments below. My recommendation is to publish this work from Communications Physics after the authors address the following issues.

Response: We thank the reviewer for appreciating our work as “well-done” and “well-written.” We respectfully disagree, however, that our work “lacks impact to satisfy high publication standard of Nature Communications.” We articulate our opinion in the point-to-point response to the reviewer’s comments. We do hope that our response and additional experiments and simulations would encourage the reviewer to be more positive regarding the impact of our work.

Comment #1: I could not agree with the authors to use “efficient” in their title because its definition is unclear. From the viewpoint of practical applications, which is emphasized in the introduction, the charge and spin conductivities are key quantities to quantify the efficiency because it determines the power consumption of magnetic devices. In this sense, it is highly desirable to evaluate the values of “ σ^z_{zx} ” and “ σ^z_{zy} ” experimentally and compare them with the other systems

mentioned in the introduction. For example, it was recently reported that the conductivity of spin current CoNiB alloy is on the order of $10^3 \text{ Ohm}^{-1} \text{ cm}^{-1}$; see Phys. Rev. Applied 14, 064056 (2020). Are the values of “ σ^z_{zx} ” and “ σ^z_{zy} ” of the present work, for example, larger than this previous work?

I do not prefer to use the spin Hall angle as an efficiency because the value can be high by using materials with high longitudinal resistivity. In addition, as I wrote above, the power consumption is determined by the conductivity, and therefore, the spin Hall angle is not a direct measure to discuss the efficiency.

As far as I know, the conductivity of Mn₃Sn is not so high compared with the metallic ferromagnets, and therefore, the bulk and interfacial generations of spin current from ferromagnets, cited as Refs. 23-32, are more “efficient” than MSHE, in my opinion. To guarantee the impact of this work and use the word “efficient” in title, I recommend the authors to specify the value of the spin conductivity and show that the value is higher than the previous proposals..

Response: First, we would like to point out that the word “efficient” in the title is used to emphasize that the critical switching current density in Mn₃Sn based heterostructure is much lower than that in the β -Ta based one. As can be seen from Fig. 4 of the original manuscript, the critical current density is reduced by almost 65%, while the switching ratio is increased by about 43%. In addition, the efficiency of switching results from exploiting the out-of-plane z -polarization of the spin current in Mn₃Sn, which requires no additional power to generate an external magnetic field to assist the switching. Based on these direct evidences acquired from our SOT switching experiments, we believe that it is appropriate to describe the MSHE-driven magnetization switching as an “efficient” process.

Yet, we do agree with the reviewer that charge and spin conductivities are key quantities to reflect the SOT efficiency. Following the reviewer’s suggestion, we have applied an AHE hysteresis loop shift method to estimate the conductivities of the y - and z - polarized out-of-plane spin currents using the method used previously by Pai *et al.*, Phys. Rev. B 93, 144409 (2016). Within this method, the AHE hysteresis loop shift ΔH_z is utilized as a direct measure of the SOT effective field H_{SOT} along the z -direction (Fig. R6a). The total effective H_{SOT} can be calculated as $H_{SOT} = (\Delta H_z(H_x = H_{x\text{sat}}) - \Delta H_z(H_x = H_{-x\text{sat}}))/2$, where the $H_{x\text{sat}}$ is the field at which ΔH_z is saturated. In our case of Mn₃Sn, both the y - and z -polarized spin currents can generate a sizable ΔH_z . Specifically, the ΔH_z generated

by the z -polarized spin current retains a finite value even in the absence of H_x (Baek *et al.*, Nat. Mater. 17, 509 (2018)), and therefore its associated effective field can be readily determined as $H_{\text{SOT}}^z = \Delta H_z (H_x = 0 \text{ Oe})$ (Liu *et al.*, Nat. Nanotechnol. 16, 277 (2021)). On the other hand, the ΔH_z generated by the y -polarized spin current only emerges with an assisted field along the x direction (H_x). Therefore, the effective SOT field generated purely by the y -polarized spin current can be estimated as $H_{\text{SOT}}^y = H_{\text{SOT}} - H_{\text{SOT}}^z$.

As an example, Fig. R6a shows the $R_{\text{AHE}}-H_z$ hysteresis curves of Mn_3Sn with $H_x = 0$ at $J = \pm 4.9 \times 10^6 \text{ A/cm}^2$ along the $[01\bar{1}0]$ direction. As discussed above, the finite shift is solely due to the z -polarized spin current. By repeating the measurements for different J , the effective SOT field per unit current density for the out-of-plane antidamping torque $\chi_{\sigma_z} = H_{\text{SOT}}^z/J_C$ is obtained by the slope in Fig. R6b, which is also indicated by the red arrow in Fig. R6c. While the slope $\chi_{\sigma_y} = H_{\text{SOT}}^y/J_C$ is contributed by the in-plane antidamping torque from the y -polarized spin current, which can also be obtained from Fig. R6c. The effective SOT efficiency (θ_{zx}^i) is consequently calculated using

$$\theta_{zx}^i = \frac{2}{\pi} \frac{2e\mu_0 M_s t}{\hbar} \chi_{\sigma_i}, \quad (1)$$

where $i = z$ or y , \hbar is the reduced Planck constant, e is the electron charge, μ_0 is the vacuum permeability, $M_s = 496 \text{ emu cm}^{-3}$ is the saturation magnetization, and $t = 1.8 \text{ nm}$ is the thickness of the Ni/Co multilayer. With the obtained $\chi_{\sigma_y} = 12.8 \times 10^{-6} \text{ Oe A}^{-1} \text{ cm}^2$ and $\chi_{\sigma_z} = 3.9 \times 10^{-6} \text{ Oe A}^{-1} \text{ cm}^2$ from Fig. R6c, θ_{zx}^y and θ_{zx}^z is determined to be 0.22 and 0.067, respectively. Since the charge conductivity of Mn_3Sn is determined to be $367.48 \mu\Omega \text{ cm}$, the spin conductivities of z - and y -polarized spin currents are $\sigma_{zx}^y = 6.02 \times 10^4 [(\hbar/2e) (\Omega \text{ m})^{-1}]$ and $\sigma_{zx}^z = 1.83 \times 10^4 [(\hbar/2e) (\Omega \text{ m})^{-1}]$, respectively, which are slightly smaller than the spin Hall conductivity in CoNiB alloy system (Hibino *et al.*, Phys. Rev. Appl. 14, 064056 (2020)), and comparable to these found in previous experiments mentioned in introduction part. A sizable ratio $|\sigma_{zx}^z/\sigma_{zx}^y| = 30.5\%$ is then estimated.

It should be noted that since the insertion of the Cu spacer layer can suppress spin currents ((Fan *et al.*, Nat. Commun. 5, 3042 (2014)), the spin conductivities of Mn_3Sn are expected to be strongly underestimated. Moreover, in our estimation for simplicity, we do not consider the enhancement of H_{SOT}^z by the reorientation of magnetic domains by H_x , which leads to the overestimation of H_{SOT}^y and

hence σ_{zx}^y . Therefore, the $|\sigma_{zx}^z/\sigma_{zx}^y|$ ratio may be even larger than that estimated above. The sizable $|\sigma_{zx}^z/\sigma_{zx}^y|$ ratio is consistent with the recent measurements in a Mn₃Sn single crystal (Kondou *et al.*, Nat. Commun. 12, 6491 (2021)).

The derived $|\sigma_{zx}^z/\sigma_{zx}^y|$ ratio, though possibly underestimated, is still sizable compared to that measured in the previous reports cited in the introduction of the main text. Moreover, there is still a room to increase the z -polarized spin current in the Mn₃Sn-based devices, by preparing a Mn₃Sn film with well-aligned magnetic domains and removing the Cu spacer layer.

Figure R6: **a** R_{AHE} vs H_z loop for Mn₃Sn based sample at $H_x = 0$ Oe with the applied current density of $\pm 4.9 \times 10^6$ A cm⁻². **b** The loop shift ΔH_z for the sample at different applied current density. **c** The slope of $\chi = \Delta H_z/J$ at different H_x values.

Following the reviewer's suggestion, we have added σ_{zx}^y and σ_{zx}^z values in the first paragraph on page 5 in the revised manuscript:

“This phenomenon cannot emerge in the conventional SOT devices where ΔH_z can only be generated by a y -polarized spin current in the presence of an external magnetic field along the x direction (H_x)³². Based on the measurements of ΔH_z with different H_x , we estimate the effective SOT fields contributed by the y -polarized and z -polarized spin currents separately, and derive the sizable spin current conductivities as $|\sigma_{zx}^y| \approx 6.02 \times 10^4$ [$(\hbar/2e)$ ($\Omega \text{ m}$)⁻¹] and $|\sigma_{zx}^z| \approx 1.83 \times 10^4$ [$(\hbar/2e)$ ($\Omega \text{ m}$)⁻¹] which are comparable to these in prior reports (Supplementary Information S5)^{22,23,29,47}. The $|\sigma_{zx}^z/\sigma_{zx}^y|$ ratio of $\sim 30.5\%$ is notable and consistent with the recently reported value for a Mn₃Sn single crystal⁴⁸, indicating the efficient generation of the z -polarized spin current in our Mn₃Sn thin films by the MSHE.”

The details of spin conductivities measurements have been included in Supplemental Information S5 with Fig. R6 added as Fig. S6. The paper that the reviewer mentioned (Hibino *et al.*, Phys. Rev. Appl. 14, 064056 (2020)) has been cited as Ref. 47 in the main text.

Comment #2: The switching experiment lacks impacts due to the following two reasons.

First, the switching probability (“portion” the authors wrote) is much lower than 100%, as can be seen in, for example, Fig. 3c. I think that the switching probability at zero H_x reflects that purely driven by MSHE. Unfortunately, the switching probability remains approximately 50%, which is too low to apply magnetic devices. The low value of the switching probability makes the superiority of MSHE, compared to the other methods mentioned in the introduction, unclear.

Response: We respectfully disagree with the reviewer’s opinion that our switching experiment lacks the impact. First, we would like to emphasize that even in the presence of the Cu spacer layer and random AFM domains, the field-free switching ratio of 50~60% in our devices is still sizable even compared to the previous reports. Moreover, the MSHE in Mn_3Sn allows us to prepare the devices using the convenient film growth direction, without specifically engineering the shape or interface of the device as done in the previous reports. As a bulk effect, the MSHE generates the spin current stronger than that originating from the non-bulk sources. Specifically, we find that the z -polarized spin current contribution is at least 30.5% of the total spin current strength, which is sizable compared to the previous reports. We also demonstrate that the switching ratio can be enhanced up to ~80% due to the reorientation of the AFM domains by a very small H_x .

Second, we would like to point out that the novelty of this work is not only in the device aspect, but also in the fundamental importance. Our experiments have demonstrated, for the first time, that the strength of the spin current and thus the SOT can be controlled via the AFM domain configuration in Mn_3Sn by a small magnetic field. This addresses the central aspect of spintronics: the control of charge-to-spin conversion.

Second, I think that not only MSHE but also SHE contributes to the switching at finite H_x . Therefore, it is necessary to evaluate the contributions from MSHE and SHE separately for guaranteeing the presence of MSHE and its importance. Again, it is necessary to evaluate the conductivity of SHE.

Response: Following the reviewer’s suggestion to compare the MSHE and SHE contributions, we

have performed additional measurements and quantitatively estimated the spin conductivities of the out-of-plane z - and in-plane y - polarized spin currents. A large $|\sigma_{zx}^z/\sigma_{zx}^y|$ ratio of $\sim 30.5\%$ was found, as described in the response to Comment #1 of this reviewer.

In order to further demonstrate the superiority of the switching by the MSHE, we have performed macro-spin simulations of a SOT device with different $\sigma_{zx}^z/\sigma_{zx}^y$ ratios. We find that the field-free switching can be realized for all nonzero $\sigma_{zx}^z/\sigma_{zx}^y$ ratios (Figs. R7 and R8). For a small $\sigma_{zx}^z/\sigma_{zx}^y$ ratio, the switching trajectory with zero H_x field is similar to the case of the conventional SOT switching with an assisted H_x field (Figs. R7a and R7b), where the magnetic moment (initially pointing along the $-z$ direction) is first pulled toward the y direction by the torque $\sim m \times (m \times y)$ exerted by the y -polarized spin current from the conventional SHE, and then relaxes to the $+z$ direction. There is no precession during the application of the current, indicating the torque is majorly $\sim m \times (m \times y)$ generated by σ_{zx}^y which directly competes with the precession. These trajectories indicate that when $\sigma_{zx}^z/\sigma_{zx}^y$ is small, the z -polarized spin current generated by the MSHE only contributes to the symmetry breaking as the assisted field does. On the other hand, when $\sigma_{zx}^z/\sigma_{zx}^y$ is not too small, the magnetic moment is directly switched to the opposite direction during the application of the current, and the precession is well maintained during the application of the current (Fig. R7c). This is a typical characteristic of the switching by the out-of-plane antidamping torque $\sim m \times (m \times z)$ due to the z -polarized spin current. Since the switching driven by the out-of-plane antidamping torque does not compete with the precession, the critical current for such switching is much smaller than that in a conventional SOT switching, and it further decreases with the increase of $\sigma_{zx}^z/\sigma_{zx}^y$ (Fig. R8).

Therefore, based on the sizable $\sigma_{zx}^z/\sigma_{zx}^y$ estimated above and the low critical switching current observed in our field-free switching, we argue that the field-free switching we observed is dominated by the out-of-plane antidamping torque driven by the z -polarized spin current resulting from the MSHE in Mn_3Sn .

Figure R7: Simulated magnetization switching trajectories with conventional y-polarized spin current switching at **a** $\sigma_{zx}^z/\sigma_{zx}^y = 0$, $H_x = 100$ Oe; and z-polarized spin current switching at **b** $\sigma_{zx}^z/\sigma_{zx}^y = 0.01$, $H_x = 0$ Oe; and **c** $\sigma_{zx}^z/\sigma_{zx}^y = 0.3$, $H_x = 0$ Oe.

Figure R8: Simulated m_z vs J switching curves for the conventional SOT switching with $\sigma_{zx}^z/\sigma_{zx}^y = 0$ at $H_x = 0$ or 100 Oe, and the z-polarized spin current switching with different $\sigma_{zx}^z/\sigma_{zx}^y$ ratios at $H_x = 0$ Oe.

We have added the following discussion in the second paragraph on page 6 of the revised manuscript:

“A remaining question is the role of the z-polarized spin current in the observed field-free switching: Whether the switching indeed occurs due to an out-of-plane antidamping torque, or the z-polarized spin current only generates a symmetry breaking perturbation to assist a conventional SOT switching by the y-polarized spin current from the SHE. To answer this question, we have performed macro-spin simulations as described in Supplementary Information S9. We find that for sizable $|\sigma_{zx}^z/\sigma_{zx}^y|$ of 30.5%, as estimated for our Mn_3Sn device, the switching dynamics exhibits typical characteristics associated with an out-of-plane antidamping torque. This indicates that the switching of our device occurs through the z-polarized spin current driven by the MSHE. On the contrary, for small $|\sigma_{zx}^z/\sigma_{zx}^y|$, we find that the switching is dominated by the y-polarized spin current due to the SHE. Importantly, our simulation results indicate that the critical current required for the MSHE-driven switching is much smaller compared to the SHE-driven switching.”

In addition, the details are included in Supplementary Information S9 with Figs. R7 and R8 added as Figs. S13 and S14.

General Comment: In summary, the work is well done and reliable. Therefore, the paper will attract readers in Nature journals. However, the work lacks enough impact to guarantee the publication from Nature Communications. Accordingly, Communications Physics would be suitable for publication.

Response: We thank the reviewer again for appreciating our work as “well done” and “reliable.” We hope that our responses to the reviewer’s comments and our additional experiments and simulations would convince the reviewer that our paper could have sufficient impact.

Minor comments.

Minor Comment #1: I recommend the authors to check their presentations of figures. For example, it is very hard to read the values of the magnetic field written in Fig. 3b because of low resolution and small font size. The unit of the conductivity in the vertical axis of Fig. S3, as well the “degree” in the horizontal axis of Fig. S3c is not shown correctly (I had downloaded the pdf version of Supplementary Information). Compared with Fig. 1d, I assumed that the unit is “ $\text{Ohm}^{-1} \text{cm}^{-1}$ ” in the horizontal axis of Fig. S3; is it correct?

Response: We thank the reviewer for pointing this out. The font size in Fig. 3b has been increased in

the revised version. Due to some word compatibility issues, the axis labels of Fig. S3c are not displayed properly. The horizontal axis of Fig. S3c (Fig. 4c in the revised version) is the angle θ between the directions of H (blue arrow) and I (red arrow), and the “degree” means degree unit. The vertical axis of Fig. S3c (Fig. 4c in the revised version) corresponds to the magnetoconductivity defined as $\Delta\sigma(H) = (\sigma(H) - \sigma(0))$, and its unit is $\Omega^{-1} \text{ cm}^{-1}$. We corrected this accordingly in the revised version.

Minor Comment #2: It would be preferable to write the pulse width of the current for switching explicitly because fast magnetization switching is required in practical applications.

Response: Following the suggestion, we have added the pulse width (800 μs) of the current used in the Methods section in the revised manuscript as follows:

“For current induced magnetization switching and anomalous Hall loop shift measurements, current pulses with an 800 μs pulse width were applied.”

Reviewer #3

General Comment: The paper is well-written, and I find all of the results to be credible and well supported by the data shown. However, in my opinion, the results shown so far do not move the field forward sufficiently to merit publication in Nature Communications. I suggest that paper be reconsidered after the authors have performed additional measurements.

Response: We thank the reviewer for appreciating our results as “credible and well supported.” We have conducted additional measurements and simulations to further strengthen our claims, and hope that the reviewer would find our revised manuscript suitable for publication in Nature Communications. Below, we respond point-to-point to the reviewer’s comments.

Comment #1: Measurement of an out-of-plane antidamping torque from a non-collinear magnet is not new. In addition to the papers cited in the manuscript under review, there are also results by Nan et al. (already in Nature Commun. 11:4671 (2020)) that the manuscript under review does not cite. That previous paper provided a quantitative measurement of the spin torque conductivity for the antidamping spin-orbit torque, which the paper under review does not provide. I do find it interesting that Mn₃Sn also provides an out-of-plane antidamping torque, but to move the field forward it is important to know how the strength of this torque compares quantitatively to previous measurements. The quantitative strength of the ordinary in-plane anti-damping spin-orbit torque should also be reported.

Response: First, we would like to point out that the paper by Nan *et al.* has been cited as Ref. 23 in the original manuscript. We agree with the reviewer that the quantitative strength of the in-plane and out-plane anti-damping SOT should be reported. Following the reviewer’s suggestion, we have applied an AHE hysteresis loop shift method to estimate the conductivities of the y- and z- polarized out-of-plane spin currents using the method used previously by Pai *et al.*, Phys. Rev. B 93, 144409 (2016). Within this method, the AHE hysteresis loop shift ΔH_z is utilized as a direct measure of the SOT effective field H_{SOT} along the z-direction (Fig. R9a). The total effective H_{SOT} can be calculated as $H_{SOT} = (\Delta H_z(H_x = H_{xsat}) - \Delta H_z(H_x = H_{-xsat}))/2$, where the H_{xsat} is the field at which ΔH_z is saturated. In our case of Mn₃Sn, both the y- and z-polarized spin currents can generate a sizable ΔH_z . Specifically, the ΔH_z generated by the z-polarized spin current retains a finite value even in the absence

of H_x (Baek *et al.*, Nat. Mater. 17, 509 (2018)), and therefore its associated effective field can be readily determined as $H_{SOT}^z = \Delta H_z (H_x = 0 \text{ Oe})$ (Liu *et al.*, Nat. Nanotechnol. 16, 277 (2021)). On the other hand, the ΔH_z generated by the y -polarized spin current only emerges with an assisted field along the x direction (H_x). Therefore, the effective SOT field generated purely by the y -polarized spin current can be estimated as $H_{SOT}^y = H_{SOT} - H_{SOT}^z$.

As an example, Fig. R9a shows the $R_{\text{AHE}}-H_z$ hysteresis curves of Mn_3Sn with $H_x = 0$ at $J = \pm 4.9 \times 10^6 \text{ A cm}^{-2}$ along the $[01\bar{1}0]$ direction. As discussed above, the finite shift is solely due to the z -polarized spin current. By repeating the measurements for different J , the effective SOT field per unit current density for the out-of-plane antidamping torque $\chi_{\sigma_z} = H_{SOT}^z/J_C$ is obtained by the slope in Fig. R9b, which is also indicated by the red arrow in Fig. R9c. While the slope $\chi_{\sigma_y} = H_{SOT}^y/J_C$ is contributed by the in-plane antidamping torque from the y -polarized spin current, which can also be obtained from Fig. R9c. The effective SOT efficiency (θ_{zx}^i) is consequently calculated using

$$\theta_{zx}^i = \frac{2}{\pi} \frac{2e\mu_0 M_s t}{\hbar} \chi_{\sigma_i}, \quad (2)$$

where $i = z$ or y , \hbar is the reduced Planck constant, e is the electron charge, μ_0 is the vacuum permeability, $M_s = 496 \text{ emu cm}^{-3}$ is the saturation magnetization, and $t = 1.8 \text{ nm}$ is the thickness of the Ni/Co multilayer. With the obtained $\chi_{\sigma_y} = 12.8 \times 10^{-6} \text{ Oe A}^{-1} \text{ cm}^2$ and $\chi_{\sigma_z} = 3.9 \times 10^{-6} \text{ Oe A}^{-1} \text{ cm}^2$ from Fig. R9c, θ_{zx}^y and θ_{zx}^z is determined to be 0.22 and 0.067, respectively. Since the charge conductivity of Mn_3Sn is determined to be $367.48 \mu\Omega \text{ cm}$, the spin conductivities of z - and y -polarized spin currents are $\sigma_{zx}^y = 6.02 \times 10^4 [(\hbar/2e) (\Omega \text{ m})^{-1}]$ and $\sigma_{zx}^z = 1.83 \times 10^4 [(\hbar/2e) (\Omega \text{ m})^{-1}]$, respectively. We find therefore a sizable ratio $|\sigma_{zx}^z/\sigma_{zx}^y| = 30.5\%$.

It should be noted that since the insertion of the Cu spacer can suppress spin currents (Fan *et al.* Nat. Commun. 5, 3042 (2014)), the spin conductivities of Mn_3Sn are expected to be strongly underestimated. Moreover, in our estimate for simplicity, we do not consider the enhancement of H_{SOT}^z by the reorientation of magnetic domains by H_x , which leads to the overestimation of H_{SOT}^y and hence σ_{zx}^y .

Therefore, the $|\sigma_{zx}^z/\sigma_{zx}^y|$ ratio may be even larger than that estimated above. The sizable $|\sigma_{zx}^z/\sigma_{zx}^y|$ ratio is consistent with the recent measurements in a Mn_3Sn single crystal (Kondou *et al.*, Nat.

Commun. 12, 6491 (2021)). This indicates the efficient generation of the z-polarized spin current by Mn₃Sn.

The derived $|\sigma_{zx}^z/\sigma_{zx}^y|$ ratio, though possibly underestimated, is still sizable compared to the values measured in previous reports (e.g., Kondou *et al.*, Nat. Commun. 12, 6491 (2021)). Moreover, there is still a room to increase the z-polarized spin current in the Mn₃Sn-based devices by preparing a Mn₃Sn film with well-aligned magnetic domains and removing the Cu spacer layer.

Figure R9: **a** R_{AHE} vs H_z loop for Mn₃Sn based sample at $H_x = 0$ Oe with the applied current density of $\pm 4.9 \times 10^6$ A cm⁻². **b** The loop shift ΔH_z for the sample at different applied current density. **c** The slope of $\chi = \Delta H_z/J$ at different H_x values.

Following the suggestion, we have estimated of the spin conductivities and torque efficiencies in the first paragraph on page 5 in the main text:

“This phenomenon cannot emerge in the conventional SOT devices where ΔH_z can only be generated by a y-polarized spin current in the presence of an external magnetic field along the x direction (H_x)³². Based on the measurements of ΔH_z with different H_x , we estimate the effective SOT fields contributed by the y-polarized and z-polarized spin currents separately, and derive the sizable spin current conductivities as $|\sigma_{zx}^y| \approx 6.02 \times 10^4$ [$(\hbar/2e)$ (Ω m)⁻¹] and $|\sigma_{zx}^z| \approx 1.83 \times 10^4$ [$(\hbar/2e)$ (Ω m)⁻¹] which are comparable to these in prior reports (Supplementary Information S5)^{22,23,29,47}. The $|\sigma_{zx}^z/\sigma_{zx}^y|$ ratio of ~30.5% is notable and consistent with the recently reported value for a Mn₃Sn single crystal⁴⁸, indicating the efficient generation of the z-polarized spin current in our Mn₃Sn thin films by the MSHE.”

The detailed discussion is included in Supplementary Information S5 with Fig. R9 added as Fig. S6.

Comment #2: Because all of the measurements are performed on samples with microns-scale lateral dimensions, the magnetic switching occurs by a process of domain wall motion. Under the influence of an applied current, this process is thermally-activated so that the primary effect of the current is heating rather than spin-orbit torque (note that the coercive magnetic field is reduced for both directions of current in Fig. 2(c) compared to Fig. 2(b), with just a small asymmetry due to spin torque as a function of the sign of the current. I realize that this is a common regime for study within the literature, but despite this fact it is not really a useful regime for obtaining any truly quantitative information about spin-orbit torques, because the thermally-activated domain-wall depinning process is poorly-controlled and difficult to model quantitatively. It is also not a useful regime for drawing any conclusions about potential applications, because practical devices will require devices much smaller in scale than 100 nm where switching is not dominated by domain-wall motion. The current densities needed to drive switching on sub-100-nm scale devices can be a factor of 50 or more greater than for devices on the many-microns scale because the switching occurs by completely different mechanisms in these two regimes (see C. Zhang et al., Appl. Phys. Lett. 107, 012401 (2015)). To show that the out-of-plane antidamping torque is strong enough for practical applications, I suggest that the authors should show that it can assist switching for devices close to the 100 nm scale or below, and not just for microns-scale devices. If the data in Fig. 2d really do indicate a full anti-damping transition driven by the current, it appears that the current densities needed to switch 100 nm scale devices might be achievable.

Response: Following this suggestion, we have fabricated the nanowire devices with a width of 100 nm. As shown in Fig. R10a, the nanowire device exhibits a perpendicular magnetic anisotropy (PMA), as seen from the nearly square loop of the $R_{\text{AHE}}-H_z$ curve. However, we have not been able to realize the field-free switching in this device (Fig. R10b). We consider that there are two possible reasons:

- 1) Due to the scaling of the width, the coercivity and anisotropy of the Ni/Co multilayer increases dramatically. Specifically, the coercivity rises from 109 Oe in the microwire to 561 Oe in the nanowire. This leads to a much larger critical current density required for switching (Zhang *et al.*, Appl. Phys. Lett. 107, 012401 (2015) and Jinnai *et al.* Appl. Phys. Lett. 113, 212403 (2018)).
- 2) With the much smaller size of the device, the inevitable Joule heating at a larger current strongly

suppresses the imbalance of the magnetic domains, resulting in a small net z -polarized spin current, which is not sufficient for switching.

On the other hand, a clear switching is still observed when an external field as small as $H_x = 5$ Oe is applied, due to the reorientation of the magnetic domains by H_x . This indicates that a field free switching in a Mn_3Sn based nanoscale SOT device can be eventually realized if the magnetic domains of Mn_3Sn are well orientated and pinned under the application of current. This may be realized by depositing Mn_3Sn on a hard ferromagnetic substrate with an in-plane anisotropy. In this case, due to a small but nonvanishing net magnetic moment of Mn_3Sn , the magnetic domains are supposed to be aligned by the strong interfacial exchange bias field. Another direction to solve this problem is to remove the Cu spacer which suppresses the spin current in Mn_3Sn based SOT devices. These aspects are beyond the scope of current work, and will be investigated elsewhere.

Figure R10: **a** Anomalous Hall loop of the 100 nm nanowire device. **b** SOT switching curves with and without the assistive field along x -axis at 300 K.

Comment #3: Another important question that would move the field forward, but the manuscript under review does not address, is whether in the switching they observe the out-of-plane antidamping torque merely provides some symmetry breaking so that the main driver of the switching is still the ordinary (inefficient) in-plane anti-damping torque with a positive overall effective damping, or whether the out-of-plane antidamping torque is strong enough to drive the more-efficient anti-damping switching mechanism.

Response: We thank the reviewer for bringing this very important aspect up. To address this comment,

we first analyze the difference between the switching of perpendicular magnetization driven by the y - or z -polarized spin currents. In case of an ordinary in-plane damping-like torque, the magnetic moment (initially pointing along the $-z$ direction) is first pulled toward the y -direction by the torque $\sim m \times (m \times y)$ from the conventional SHE, and then relaxes to the $+z$ direction by the symmetry breaking perturbation when the current is released. There is no precession during the application of the current since the torque directly competes with the precession. On the other hand, in case of the out-of-plane antidamping torque, the magnetic moment is directly switched to the opposite direction even without the assistance of any symmetry breaking perturbation. Since the torque $\sim m \times (m \times z)$ from the z -polarized spin current does not compete with the precession, the precession is well maintained during the application of the current, and the critical current for such switching is much smaller than that in a conventional SOT switching.

Figure R11: Simulated magnetization switching trajectories with conventional y -polarized spin current switching at **a** $\sigma_{zx}^z/\sigma_{zx}^y = 0$, $H_x = 100$ Oe; and z -polarized spin current switching at **b** $\sigma_{zx}^z/\sigma_{zx}^y = 0.01$, $H_x = 0$ Oe; and **c** $\sigma_{zx}^z/\sigma_{zx}^y = 0.3$, $H_x = 0$ Oe.

In order to further demonstrate the superiority of the switching by the MSHE, we have performed macro-spin simulations of a SOT device with different $\sigma_{zx}^z/\sigma_{zx}^y$ ratios. We find that the field-free switching can be realized for all nonzero $\sigma_{zx}^z/\sigma_{zx}^y$ ratios (Figs. R11 and R12). For a small $\sigma_{zx}^z/\sigma_{zx}^y$ ratio,

the switching trajectory with zero H_x field is similar to the case of the conventional SOT switching with an assisted H_x field (Figs. R11a and R11b), where the magnetic moment (initially pointing along the $-z$ direction) is first pulled toward the y direction by the torque $\sim m \times (m \times y)$ exerted by the y -polarized spin current from the conventional SHE, and then relaxes to the $+z$ direction. There is no precession during the application of the current, indicating the torque is majorly $\sim m \times (m \times y)$ generated by σ_{zx}^y which directly competes with the precession. These trajectories indicate that when $\sigma_{zx}^z/\sigma_{zx}^y$ is small, the z -polarized spin current generated by the MSHE only contributes to the symmetry breaking as the assisted field does. On the other hand, when $\sigma_{zx}^z/\sigma_{zx}^y$ is not too small, the magnetic moment is directly switched to the opposite direction during the application of the current, and the precession is well maintained during the application of the current (Fig. R11c). This is a typical characteristic of the switching by the out-of-plane antidamping torque $\sim m \times (m \times z)$ due to the z -polarized spin current. Since the switching driven by the out-of-plane antidamping torque does not compete with the precession, the critical current for such switching is much smaller than that in a conventional SOT switching, and it further decreases with the increase of $\sigma_{zx}^z/\sigma_{zx}^y$ (Fig. R12).

Figure R12: Simulated m_z vs J switching curves for the conventional SOT switching with $\sigma_{zx}^z/\sigma_{zx}^y = 0$ at $H_x = 0$ or 100 Oe, and the z -polarized spin current switching with different $\sigma_{zx}^z/\sigma_{zx}^y$ ratios at $H_x =$

0 Oe.

In addition, we have performed new measurements to quantitatively estimate the spin Hall conductivity σ_{zx}^y and the magnetic spin Hall conductivity σ_{zx}^z and included these results in Supplementary Information S5. Based on the sizable $\sigma_{zx}^z/\sigma_{zx}^y$ ratio of 0.305 extracted and the small J_c observed, we can deduce that the field-free switching in this work is predominantly realized by the out-of-plane antidamping torque, which is contributed by the z -polarized spin current from MSHE in Mn_3Sn .

In this revision, we have added this discussion in the second paragraph of page 6:

“A remaining question is the role of the z -polarized spin current in the observed field-free switching: Whether the switching indeed occurs due to an out-of-plane antidamping torque, or the z -polarized spin current only generates a symmetry breaking perturbation to assist a conventional SOT switching by the y -polarized spin current from the SHE. To answer this question, we have performed macro-spin simulations as described in Supplementary Information S9. We find that for sizable $|\sigma_{zx}^z/\sigma_{zx}^y|$ of 30.5%, as estimated for our Mn_3Sn device, the switching dynamics exhibits typical characteristics associated with an out-of-plane antidamping torque. This indicates that the switching of our device occurs through the z -polarized spin current driven by the MSHE. On the contrary, for small $|\sigma_{zx}^z/\sigma_{zx}^y|$, we find that the switching is dominated by the y -polarized spin current due to the SHE. Importantly, our simulation results indicate that the critical current required for the MSHE-driven switching is much smaller compared to the SHE-driven switching.”

The details on macro-spin simulation results are added in Supplemental Information S9 and the Fig. R11 and R12 are added as Fig. S13 and S14.

Comment #4: The paper does not appear to provide all of the materials parameters that readers will want to know for a full analysis of the results, for example the resistivities of all of the various layers and the average magnetic moment per unit volume of the Ni/Co multilayer.

Response: Following the reviewer’s suggestion, we have added values of resistivity ($\rho_{Mn_3Sn} = 367.5 \mu\Omega\cdot\text{cm}$) and magnetization ($M_S = 496 \text{ emu/cm}^3$) in the second paragraph on page 4:

“The resistivities of Mn_3Sn and $Cu/[Ni/Co]_3/Cu$ are determined to be $367.5 \mu\Omega\text{-cm}$ and $45.0 \mu\Omega\text{-cm}$, respectively. The saturation magnetization of Ni/Co multilayer has been measured to be 496 emu/cm^3 .”

Comment #5: I would have found it useful to understand what is the arrangement of the spins that allow a small nonzero total magnetic moment in the Mn_3Sn . If the lower two spins shown in the diagram for AFM1 in Fig. 1(c) point exactly along the sides of the equilateral triangle as shown, the net magnetization will be zero. I realize that the canting away from this configuration will be small, but I suggest that the authors indicate this canting somehow.

Response: As described in the main text and Supplementary Information, Mn_3Sn has two antiferromagnetic alignments, *i.e.*, AFM1 and AFM2. The two alignments exhibit different symmetry. AFM1 has a mirror symmetry M_x perpendicular to the x ($[01\bar{1}0]$) direction, which reverses the y - and z - components of the magnetic moments but does not influence the x -component. Therefore, as shown in Fig. R13, the Mn atoms on the different side of the M_x plane have the same x -component of the moment and opposite y - and z -components. Although in the ideal alignment, the angle between two moments must be 120° , small canting is allowed in reality, since the canted alignment still fulfils the requirement of the M_x symmetry (Fig. R13, AFM1). This canting produces a small but nonvanishing in-plane net magnetization along the x -direction. Similarly, AFM2 has glide symmetry $G_y = \{M_y | \frac{c}{2}\}$ (a mirror operation plus a half-unit-cell translation along the $[0001]$ direction) perpendicular to the y ($[2\bar{1}\bar{1}0]$) direction, which allows canting resulting in a small but nonvanishing in-plane net magnetization along y -direction (Fig. R13, AFM2).

Figure R13: Illustration of the Mn_3Sn spin configuration indicating spin canting. The solid arrows denote the magnetic moments in the ideal alignments, and the dashed arrows denote the canted

moments allowed by symmetry. This symmetry allows finite canting of magnetic moments as schematically shown in Fig. S1a, resulting in net magnetization along the x(y) direction for AFM1(2).

We have included this discussion in Supplemental Information S1 and replaced Fig. S1a with Fig. R13 with a caption:

“This symmetry allows finite canting of magnetic moments as schematically shown in Fig. S1a, resulting in net magnetization along the x(y) direction for AFM1(2).”

Comment #6: I suggest that the authors state explicitly how they define the quantity ΔH_z graphed in Fig. 2(d). For a given value of applied current, is it the difference of the coercive fields for the magnet originally in the up configuration and the down configuration? I do not believe this is ever stated explicitly.

Response: The shift ΔH_z is defined as $\Delta H_z = H_{center}(I^+) - H_{center}(I^-)$, where $H_{center}(I^\pm) = \frac{[H_r^+(I^\pm) - H_r^-(I^\pm)]}{2}$ is the center of the hysteresis loop determined by the difference of positive and negative magnetization-reversal fields $H_r^\pm(I)$, and I^\pm are positive and negative currents.

We have added this definition to the first paragraph on page 5 of the main text.

“Here we define the shift of the RAHE- H_z hysteresis loop as $\Delta H_z = H_{center}(I^+) - H_{center}(I^-)$, where $H_{center}(I^\pm) = \frac{[H_r^+(I^\pm) - H_r^-(I^\pm)]}{2}$ is the center of the hysteresis loop determined by the difference of positive and negative magnetization-reversal fields $H_r^\pm(I)$, and I^\pm are positive and negative currents.”

General Comment: On a positive note, it is interesting and exciting that the authors observe signs of an out-of-plane antidamping torque from a new material (Mn₃Sn), and I find it really neat that they are able to control the overall effectiveness of this torque by applying a magnetic field to change the arrangement of antiferromagnetic domains. However, to merit publication in Nature Communications it is my opinion that the work should move the field forward more significantly than this. I suggest the authors perform additional measurements to address the questions noted in my comments 1-3 above. If they can address a significant portion of these comments I would likely recommend the paper for publication in Nature Communications.

Response: We thank again the reviewer for the positive and insightful comments. We believe our additional measurements and simulations have addressed the reviewer's concerns, and hope that the reviewer would find our revised manuscript suitable for publication in Nature Communications.

REVIEWER COMMENTS

Reviewer #1 (Remarks to the Author):

Authors addressed most of issues.

One remaining issue is the stability of Mn₃Sn spin texture when the current is applied. From the response letter, the author give the following information:

T_N of Mn₃Sn is 420K.

The pulse width is 200us used for Rxx measurement during temperature calibration, however, the SOT was performed at the pulse width of 800 us. Therefore, the estimated temperature for the SOT switching (T=360 K) and the loop shift measurement (T=340K) are lower than the real cases.

Authors should estimate the temperature again.

In addition, when the pulse width is reduced to ns, can the same results be observed? That is, how important is the thermally assisted switching?

Furthermore, would the spin texture be significantly disturbed even at 360 K so that spin texture cannot still be hold to give rise to the z-polarized spin current? Although the device temperature is lower than the Neel or blocking temperature, at higher temperature the stability of spin texture can be reduced. Authors should comment on it.

Reviewer #2 (Remarks to the Author):

First of all, I would like to express my deepest gratitude to the authors for sincerely replying my previous comments. I understand that the authors have made great effort for making replies.

However, I still do not think that this work satisfies high publication standards from Nature Communications. For example, the authors replied that the switching probability, 50 %, is still sizable even compared to the previous reports. I do not prefer this reply. In my opinion, the publication policy of Nature Communications is not based on such a relativ, and minor in this case, progress; rather, Nature Communications require great advance in the research field. Moreover, I still do not think that the present work has an impact overcoming previous works published from Nature group, such as the switching experiments by Baek, K.-J. Lee, and Stiles (2018). My recommendation is unchanged, i.e., transferring to Communications Physics. No more review will be necessary.

Reviewer #4 (Remarks to the Author):

In their paper "efficient perpendicular magnetization switching by a magnetic spin Hall effect in a noncollinear antiferromagnet", the authors Shuai Hu et al. report of switching of the magnetization of a perpendicularly magnetized Ni/Co multilayer separated by a thin Cu layer from a spin current source layer, in this case a thin film of Mn₃Sn. Their findings hint to switching by the magnetic spin Hall effect (MSHE) and not from the bulk SHE exploiting a spin current with perpendicular spin polarization. These findings are interesting in the context of switching spintronic devices since the authors demonstrate deterministic switching in zero magnetic field and believe that the switching efficiency is superior to standard SOT based switching schemes.

As a reviewer of only the second round of reviews, I would like to acknowledge that the authors responded adequately to the requests of the three reviewers and went through quite an effort to improve the paper. In particular, they provided (as required by the reviewers) additional measurements which are partially included in the new version of the supplementary material.

I have two points that I would like to stress.

First, I think that the data provided by the authors in the response to the reviewers concerning switching of a nanostructured element (Fig. R10) should be included in the

paper. I believe it is highly relevant information for researchers in the field, that field free switching is not achieved in such a device! Furthermore the data presented in Fig. R10b) shows that also in structures as small as 100 nm switching is only partial. Indeed, the observed AHE signal is only a fraction of the reported signal in Fig. R 10a. The author should comment on this and include this data at least in the supplementary material. Second, I do not think that the presented macrospin simulations can represent the physical processes underlying the switching process. First of all, the significant temperature increase is not reflected in the simulations. The author show that the temperature increase reaches up to 360K, thus temperature effects need to be taken into account. Actually, the authors fail to report the Neel temperature of their thin films as requested by reviewer1. Furthermore, the pulse length in the experiments is significantly longer than in the simulations (leading again to significant heating).

In summary, I find the paper interesting and a possible paper for Nature Communications if the points mentioned above a clearly addressed.

Response to Reviewers

Reviewer #1:

General Comment: Authors addressed most of issues.

Response: We sincerely thank the reviewer to find most of issues have been satisfactorily addressed. Below, we respond point-to-point to the reviewer's comments.

Comment #1: One remaining issue is the stability of Mn₃Sn spin texture when the current is applied.

From the response letter, the author give the following information:

T_N of Mn₃Sn is 420K.

The pulse width is 200us used for R_{xx} measurement during temperature calibration, however, the SOT was performed at the pulse width of 800 us. Therefore, the estimated temperature for the SOT switching (T=360 K) and the loop shift measurement (T=340K) are lower than the real cases.

Authors should estimate the temperature again.

Response: We thank the reviewer for raising the concern. The pulse width of 200 μs indicated in Fig. R1 in our previous response letter was a misprint. The pulse width has been consistently set as 800 μs throughout this study, including both the R_{xx} measurement and the SOT switching measurement.

Following the reviewer's suggestion, we have performed additional temperature calibration of the sample and confirmed our consistent estimation on the Joule heating effect. In the revised Supplementary Information, the current pulse width of 800 μs for R_{xx} measurements during temperature calibration has been stated in the Fig. S10 caption.

Comment #2: In addition, when the pulse width is reduced to ns, can the same results be observed? That is, how important is the thermally assisted switching?

Response: We agree with the reviewer that the role of thermally assisted switching should be examined carefully. In response to his/her comment, we have conducted additional macro-spin simulations where we varied the pulse widths from 1 to 1000 ns. In these simulations, a Gaussian-distributed random thermal fluctuation field H_{th} with mean = 0 and standard deviation = $\sqrt{\frac{2\alpha k_B T}{\gamma M_s V \delta t}}$ was included to account

for the temperature effect. Here k_B is the Boltzmann constant, T is the temperature, V is the volume of a ferromagnet taken to be $50 \text{ nm} \times 50 \text{ nm} \times 1.8 \text{ nm}$, γ is the gyromagnetic ratio, and δt is the integration time step [D. K. Lee et al. Sci Rep 10, 1772 (2020)]. For a pure y-polarized spin current, i.e. $\sigma_{zx}^z/\sigma_{zx}^y = 0$, we found that the field-free switching was never archived even with a strong thermal assistance. We then compared the switching in the presence of either finite in-plane field H_x or finite σ_{zx}^z . As an example, we show in Fig. R1 the switching trajectory in the presence of a long pulse width of 1000 ns. It is seen that the inclusion of a thermal fluctuation field does not produce any qualitative influence on the magnetization switching trajectory compared to our previous simulation (Fig. S15) except additional noises. The magnetization switching trajectory with other tested pulse widths demonstrated similar behavior.

Fig. R1 Simulated magnetization switching trajectories in the presence of the thermal fluctuation field with conventional y-polarized spin current switching **a** and **b** $\sigma_{zx}^z/\sigma_{zx}^y = 0$, $H_x = 100 \text{ Oe}$ at 300 and 370 K, respectively; and z-polarized spin current switching **c** and **d** $\sigma_{zx}^z/\sigma_{zx}^y = 0.01$, $H_x = 0 \text{ Oe}$ at 300 and 370 K, respectively; **e** and **f** $\sigma_{zx}^z/\sigma_{zx}^y = 0.3$, $H_x = 0 \text{ Oe}$ at 300 K and 370 K, respectively.

We have also simulated the critical switching current with different pulse widths for $\sigma_{zx}^z/\sigma_{zx}^y = 0$, $H_x = 100$ Oe (Fig. R2a) and $\sigma_{zx}^z/\sigma_{zx}^y = 0.3$, $H_x = 0$ Oe (Fig. R2b) at 300 K and 370 K. In both cases, the switching current density increased at short pulses, similar to the previous work [K.S. Lee et al. Appl. Phys. Lett. 104, 072413 (2014)]. Moreover, in agreement with the reviewer's expectation, the thermally assisted switching played a more important role in the case of the z -polarized spin current than in the case of the y -polarized current, as is evident from the larger increase of the critical current density with the decreasing pulse width. We note, however, that, in terms of the switching efficiency, the z -polarized spin current is still superior than the y -polarized spin current due to a much lower current amplitude required for switching in the entire simulated range. In addition, by comparing the results at 300 K and 370 K, the almost overlapped curves in Fig. R2b suggest that the z -polarized spin current is more immune to the temperature increase induced by Joule heating. Based on this additional analysis, we conclude that neither the reduced pulse width nor the thermal effects impact our main result emphasizing the efficient field-free switching of the perpendicular magnetization driven by the z -polarized spin current generated by a magnetic spin Hall effect.

Fig. R2 Simulated critical current density at 300 K and 370 K with **a** conventional y -polarized spin current switching ($\sigma_{zx}^z/\sigma_{zx}^y = 0$, $H_x = 100$ Oe) and **b** z -polarized spin current switching ($\sigma_{zx}^z/\sigma_{zx}^y = 0.3$, $H_x = 0$ Oe).

In this revision, we have added the above discussion in section S11 of Supplementary Information and included Figs. R1 and R2 as Figs. S17 and S18.

Comment #3: Furthermore, would the spin texture be significantly disturbed even at 360 K so that spin

texture cannot still be hold to give rise to the z-polarized spin current? Although the device temperature is lower than the Neel or blocking temperature, at higher temperature the stability of spin texture can be reduced. Authors should comment on it.

Response: We thank the reviewer for this comment. Following the reviewer's suggestion, we have first evaluated the temperature influence on the spin torque by measuring the harmonic anomalous Hall loops at elevated temperatures. It is well established that the peak amplitude of second harmonic anomalous Hall loop is proportional to the strength of spin torque. The results are shown in Fig. R3. The first harmonic anomalous Hall loops in Fig. R3a demonstrate good perpendicular magnetic anisotropy of the device at various temperatures. On other hand, in Fig. R3b, one can see that the shape of second harmonic Hall loop is largely preserved at around 360 K. Upon a further increase of temperature toward 420 K, the second harmonic Hall loop shows much smaller signal compared to the one at 298 K. The results in Fig. R3 indicate that the Mn_3Sn spin texture and its associated spin torque are maintained in our device during the measurement. Upon quenching of spin texture by elevating temperature toward the Mn_3Sn Neel temperature, the spin torque generated by Mn_3Sn is largely suppressed.

Following the suggestion and above discussion, we have added the temperature dependence of spin torque to Supplementary Information S8 with Fig. R3 included as Fig. S11.

Figure R3: The first harmonic (a) and second harmonic (b) anomalous Hall loops measured at various temperature for the Mn₃Sn sample. An ac current with a current density of 1×10^7 A/cm² and a frequency of 13.7 Hz is employed for the measurements. The actual device temperature T_{actual} is determined through temperature dependence of device resistance and indicated beside each second harmonic anomalous Hall loop in b.

Reviewer #2

Comment #1: First of all, I would like to express my deepest gratitude to the authors for sincerely replying my previous comments. I understand that the authors have made great effort for making replies. However, I still do not think that this work satisfies high publication standards from Nature Communications. For example, the authors replied that the switching probability, 50 %, is still sizable even compared to the previous reports. I do not prefer this reply. In my opinion, the publication policy of Nature Communications is not based on such a relativ, and minor in this case, progress; rather, Nature Communications require great advance in the research field. Moreover, I still do not think that the present work has an impact overcoming previous works published from Nature group, such as the switching experiments by Baek, K.-J. Lee, and Stiles (2018). My recommendation is unchanged, i.e., transferring to Communications Physics. No more review will be necessary.

Response: We thank the reviewer for his/her appreciation of our efforts. However, it is unfortunate that the reviewer does not recognize the importance of our work. We would like to emphasize again our key advances and conceptual novelties: 1) We show, for the first time, that the polarization and the strength of the MSHE-generated spin current and thus the SOT can be controlled via the antiferromagnetic domain configuration in Mn_3Sn by a magnetic field; 2) For the first time, we demonstrate the reversible polarity of the SOT-induced magnetization switching originating from the MSHE in Mn_3Sn . The MSHE-controlled SOT is fundamentally not possible with conventional heavy metals due to the lack of a magnetic order in these materials; 3) We find that the SOT generated by the MSHE is much more efficient for switching a perpendicular-magnetized ferromagnet than that resulting from the conventional spin Hall effect, due to the much smaller switching current and the absence of requirement of an assisted magnetic field and. The enhanced efficiency originates from the unconventional SOT driven by the MSHE-induced out-of-plane polarized spin current. We are confident that these results have broad interest and potentially strong impact and thus warrant publication of our work in Nature Communications.

Reviewer #4

General Comment: In their paper "efficient perpendicular magnetization switching by a magnetic spin Hall effect in a noncollinear antiferromagnet", the authors Shuai Hu et al. report of switching of the magnetization of a perpendicularly magnetized Ni/Co multilayer separated by a thin Cu layer from a spin current source layer, in this case a thin film of Mn₃Sn. Their findings hint to switching by the magnetic spin Hall effect (MSHE) and not from the bulk SHE exploiting a spin current with perpendicular spin polarization. These findings are interesting in the context of switching spintronic devices since the authors demonstrate deterministic switching in zero magnetic field and believe that the switching efficiency is superior to standard SOT based switching schemes.

As a reviewer of only the second round of reviews, I would like to acknowledge that the authors responded adequately to the requests of the three reviewers and went through quite an effort to improve the paper. In particular, they provided (as required by the reviewers) additional measurements which are partially included in the new version of the supplementary material.

I have two points that I would like to stress.

Response: We are grateful to the reviewer for his/her recognition on the significance of our work. Below we made a point-to-point response for the reviewer's comments.

Comment #1: First, I think that the data provided by the authors in the response to the reviewers concerning switching of a nanostructured element (Fig. R10) should be included in the paper. I believe it is highly relevant information for researchers in the field, that field free switching is not achieved in such a device! Furthermore the data presented in Fig. R10b) shows that also in structures as small as 100 nm switching is only partial. Indeed, the observed AHE signal is only a fraction of the reported signal in Fig. R 10a. The author should comment on this and include this data at least in the supplementary material.

Response: We thank the reviewer for this comment. Following the reviewer's suggestion, we have included the results of our nanowire device switching experiments and the associated discussion in Supplementary Information S9. We have also commented there on the possible origin of the partial domain switching induced by MSHE as compared to the switching by an applied magnetic field.

As mentioned in the response letter of the first round, the dramatic increase of coercivity and

anisotropy of Co/Ni and the non-uniform Mn₃Sn domain due to Joule heating at a larger current could be responsible for the partial switching at the nanoscale. We argue that a field free switching in a Mn₃Sn based nanoscale SOT device can be eventually realized if the magnetic domains of Mn₃Sn are well orientated and pinned under the application of current. This may be realized by depositing Mn₃Sn on a hard ferromagnetic substrate with an in-plane anisotropy. In this case, due to a small but nonvanishing net magnetic moment of Mn₃Sn, the magnetic domains are supposed to be aligned by the strong interfacial exchange bias field. Another direction to solve this problem is to remove the Cu spacer which suppresses the spin current in Mn₃Sn based SOT devices. These aspects are beyond the scope of current work, and will be investigated elsewhere. In this revision, we have added the nanowire switching results and the discussions above in the Supplementary Information S9.

Comment #2: Second, I do not think that the presented macrospin simulations can represent the physical processes underlying the switching process. First of all, the significant temperature increase is not reflected in the simulations. The author show that the temperature increase reaches up to 360K, thus temperature effects need to be taken into account. Actually, the authors fail to report the Neel temperature of their thin films as requested by reviewer1. Furthermore, the pulse length in the experiments is significantly longer than in the simulations (leading again to significant heating).

Response: We agree with the reviewer on the importance of temperature effect. Therefore, we have first evaluated the temperature influence on the spin torque by measuring the harmonic anomalous Hall loops at elevated temperatures. It is well established that the peak amplitude of second harmonic anomalous Hall loop is proportional to the strength of spin torque. The results are shown in Fig. R4. The first harmonic anomalous Hall loops in Fig. R4a demonstrate good perpendicular magnetic anisotropy of the device at various temperatures. On other hand, in Fig. R4b, one can see that the shape of second harmonic Hall loop is largely preserved at around 360 K. Upon a further increase of temperature toward 420 K, the second harmonic Hall loop shows much smaller signal compared to the one at 298 K. The results in Fig. R4 indicate that the Mn₃Sn spin texture and its associated spin torque are maintained in our device during the measurement. Upon quenching of spin texture by elevating temperature toward the Mn₃Sn Neel temperature, the spin torque generated by Mn₃Sn is largely suppressed.

Figure R4: The first harmonic (a) and second harmonic (b) anomalous Hall loops measured at various temperature for the Mn_3Sn sample. An ac current with a current density of $1 \times 10^7 \text{ A/cm}^2$ and a frequency of 13.7 Hz is employed for the measurements. The actual device temperature T_{actual} is determined through temperature dependence of device resistance and indicated beside each second harmonic anomalous Hall loop in b.

Secondly, in this revision, we have added additional micromagnetic simulation where we varied the pulse widths from 1 to 1000 ns and considered the thermal effects. To do so, a Gaussian-distributed random thermal fluctuation field H_{th} with mean = 0 and standard deviation = $\sqrt{\frac{2\alpha k_B T}{\gamma M_s V \delta t}}$ was added into the original LLG equation, where k_B is the Boltzmann constant, T is the temperature, V is the volume of FM taken as $50 \text{ nm} \times 50 \text{ nm} \times 1.8 \text{ nm}$, γ is the gyromagnetic ratio, and δt is the integration time step [Lee, DK., Lee, KJ. Sci Rep 10, 1772 (2020)]. First, for a pure y-polarized spin current, *i.e.* $\sigma_{zx}^z / \sigma_{zx}^y = 0$, we find that the switching is never achieved in the absence of an in-plane assisted magnetic field H_x , even with a strong thermal assistance. This indicates that the thermal effect

alone cannot result in a field-free switching of the perpendicular magnetization. Second, we compare the switching behaviors at 300 K and 370 K in the presence of either finite in-plane field H_x or finite σ_{zx}^z for pulse widths in the range of 1 ns to 1000 ns. As an example, we show in Fig. R5 the switching trajectory in the presence of a long pulse width of 1000 ns. Except additional noises present, including the thermal fluctuation field does not introduce qualitative influence field does not produce any qualitative effect on the magnetization switching trajectory compared to previous simulation (Fig. S15). The magnetization switching trajectory with other tested pulse widths demonstrate similar behavior.

Figure R5: Simulated magnetization switching trajectories after considering the thermal fluctuation field with conventional y-polarized spin current switching (a) and (b) $\sigma_{zx}^z/\sigma_{zx}^y = 0$, $H_x = 100$ Oe at 300 K and 370 K, respectively; and z-polarized spin current switching (c) and (d) $\sigma_{zx}^z/\sigma_{zx}^y = 0.01$, $H_x = 0$ Oe at 300 K and 370 K, respectively; (e) and (f) $\sigma_{zx}^z/\sigma_{zx}^y = 0.3$, $H_x = 0$ Oe at 300 K and 370 K, respectively.

Fig. R6 summarizes the critical switching current with different pulse widths for $\sigma_{zx}^z/\sigma_{zx}^y = 0$, $H_x = 100$ Oe (Fig. R6a) and $\sigma_{zx}^z/\sigma_{zx}^y = 0.3$, $H_x = 0$ Oe (Fig. R6b) at 300 K and 370 K, respectively. In both cases, the switching current would increase at short pulses, similar to previous work [K. S. Lee et al. Appl. Phys. Lett. 104, 072413 (2014)]. Moreover, we find that the thermally assisted switching plays a more important role in the case of z -polarized spin current than that of y -polarized one, as manifested by the larger increase of critical current with the decrease of pulse width. However, it should be noted that, in terms of the switching efficiency, the z -polarized case is still superior than y -polarized one: a much lower current amplitude is required in the entire simulated range. In addition, by comparing the results at 300 K and 370 K, the almost overlapped curve in the z -polarized case also suggest that it is more immune to the temperature increase by Joule heating. Based on these additional examinations, we can conclude that the thermal effect would not affect our main observations, *i.e.* the field-free and efficient switching of the perpendicular magnetization is majorly due to the z -polarized spin current generated by a magnetic spin Hall effect.

Figure R6: Simulated critical current density with conventional y -polarized spin current switching with (a) $\sigma_{zx}^z/\sigma_{zx}^y = 0$, $H_x = 100$ Oe at 300 K and 370 K; and z -polarized spin current switching with (b) $\sigma_{zx}^z/\sigma_{zx}^y = 0.01$, $H_x = 0$ Oe; and (c) $\sigma_{zx}^z/\sigma_{zx}^y = 0.3$, $H_x = 0$ Oe at 300 K and 370 K.

In this revision, we have added the temperature dependence of spin torque to Supplementary Information S8 with Fig. R4 included as Fig. S11. The micromagnetic simulations with varied pulse width and temperature have been added to section S11 in SI.

Comment #3: In summary, I find the paper interesting and a possible paper for Nature Communications if the points mentioned above a clearly addressed.

Response: We sincerely thank the reviewer for the recommendation. We hope that our responses and revisions have addressed the remaining concerns of this reviewer and the other reviewers.

REVIEWERS' COMMENTS

Reviewer #1 (Remarks to the Author):

All the questions I raised have been carefully addressed by the authors.

Response to Reviewers

Reviewer #1 (Remarks to the Author):

General Comments: All the questions I raised have been carefully addressed by the authors.

Response: We thank the reviewer for taking efforts to review and recommend the acceptance of our work.